# No detectable Weddell Sea Antarctic Bottom Water export during the Last and Penultimate Glacial Maximum

Huang Huang [1]*, Marcus Gutjahr [1], Anton Eisenhauer [1] & Gerhard Kuhn [2]

Weddell Sea-derived Antarctic Bottom Water (AABW) is one of the most important deep water masses in the Southern Hemisphere occupying large portions of the deep Southern Ocean (SO) today. While substantial changes in SO-overturning circulation were previously suggested, the state of Weddell Sea AABW export during glacial climates remains poorly understood. Here we report seawater-derived Nd and Pb isotope records that provide evidence for the absence of Weddell Sea-derived AABW in the Atlantic sector of the SO during the last two glacial maxima. Increasing delivery of Antarctic Pb to regions outside the Weddell Sea traced SO frontal displacements during both glacial terminations. The export of Weddell Sea-derived AABW resumed late during glacial terminations, coinciding with the last major atmospheric $CO_2$ rise in the transition to the Holocene and the Eemian. Our new records lend strong support for a previously inferred AABW overturning stagnation event during the peak Eemian interglacial.

[1] GEOMAR Helmholtz Centre for Ocean Research Kiel, Wischhofstraße 1-3, 24148 Kiel, Germany. [2] Alfred-Wegener-Institut Helmholtz-Zentrum für Polar- und Meeresforschung, Am Alten Hafen 26, 27568 Bremerhaven, Germany. *email: huang17323@gmail.com

The Southern Ocean (SO) has long been identified as the key driver in regulating atmospheric $CO_2$ concentrations on glacial–interglacial timescales, mediating ocean–atmosphere carbon dioxide exchange by the interaction of variable export production[1] and circulation changes[2–5]. Dissolved inorganic carbon (DIC)-enriched deep waters were increasingly upwelled to the SO surface during glacial terminations, where this DIC is degassed to the atmosphere as $CO_2$. During the last deglaciation (~18–11 ka BP), atmospheric $pCO_2$ increased during two main intervals between 18–15 ka and 13–11 ka[6], which occurred during Southern Hemispheric warming phases[7]. The multi-millennial-scale deglacial atmospheric $pCO_2$ rise was largely a consequence of a southward-moving SO-overturning cell[2,8–10] and the associated increased upwelling in the Southern Ocean. At least three intervals of centennial-scale atmospheric $pCO_2$ rise were reported during Termination I (at 16.3, 14.8, and 11.7 ka) that can be linked to Northern Hemispheric climate transitions, likely controlled by AMOC reinvigoration and the associated climatic teleconnections[11].

In contrast to the state of knowledge regarding the understanding of deglacial frontal shifts and increasing deep SO overturning, strikingly little is known with respect to changes in water mass sourcing before and during glacial terminations. For example, while deep SO water masses at the end of the Last Glacial Maximum (LGM) were identified as being depleted in both $\delta^{13}C$[12] and radiocarbon[3,5] alongside relatively low dissolved oxygen concentrations[4], the geographic origin of this glacial deep SO water mass is unresolved to date. This situation is dissatisfactory since commonly termed Southern Source Waters (SSW) have been invoked to occupy large parts of the deeper Atlantic Ocean at the expense of North Atlantic Deep Water (NADW) during the Last Glacial Maximum and major Northern Hemisphere Stadials[13,14], with consequences for deep ocean carbon sequestration[15]. No study could as yet unambiguously demonstrate that this postulated SSW is the glacial equivalent of interglacial Antarctic Bottom Water or whether SSW originated from other abyssal parts of the world's oceans.

Weddell Sea-derived AABW is formed at several locations on the Antarctic shelf today, most prominently in front of the Filchner, Ronne, and Larsen ice shelfs in the southern and western Weddell Sea[16,17]. Weddell Sea Bottom Water is the coldest and densest variety of Antarctic Bottom Water formed within the Weddell Sea[18]. It mixes with the overlying Weddell Sea Deep Water and modified warm deep water that is advected into the Weddell Sea within the Weddell Gyre, eventually leaving the basin to the north and northeast as Weddell Sea Deep Water[19–21], which we refer to as Weddell Sea-derived AABW below. Within the Scotia Sea, freshly exported Weddell Sea-derived AABW mixes with Lower Circumpolar Deep Water arriving from the Pacific sector of the SO before its advection to our sediment core sites in the Atlantic sector of the SO[22].

We address the issue of SO water mass sourcing by using two sensitive provenance proxies, the authigenic Fe–Mn oxyhydroxide-bound neodymium (Nd) and lead (Pb) isotope signatures in deep SO sediments. Both trace metals are supplied to the oceans mainly from continental runoff or dust input[23]. Yet while Nd is also supplied via porewater exchange processes along continental margins[24], such an input mechanism has as yet not been reported for Pb. Both proxies carry the isotopic composition of their continental source area and subsequent alteration through water mass mixing along their flow path. Since Pb is more particle-reactive than Nd, its residence time in seawater is significantly shorter on the order of few tens of years[25], compared with several hundreds of years for dissolved Nd[26]. The high particle reactivity of Pb also leads to its special behavior in seawater in contrast to Nd. A recent study[27] demonstrated that even

without vertical water mass mixing, temporally changing Pb isotopic compositions in North Pacific surface water left a footprint in the underlying deep-water Pb isotope signal through seawater-particle exchange by sinking particles. Consequently, the seawater-derived authigenic Pb isotope signature in deep marine sediments in the Southern Ocean can be controlled both by changes in the upper and lower overturning cells. In contrast to authigenic Pb, the Nd isotope signal of the same deep marine authigenic Fe–Mn oxyhydroxide fraction only records bottom or porewater compositions[28,29]. Taking advantage of the different behavior of Pb and Nd in seawater, past seawater Pb isotopic compositions extracted from the authigenic Fe–Mn oxyhydroxide fraction in deep marine sediments[30] hence provide insights into isotopic changes of the entire water column, while the Nd isotope composition extracted from deep marine sediments is controlled by bottom water conditions.

Our records resolve changes in the upper and lower SO-overturning cells[18,31] across the two most recent glacial–interglacial transitions (Termination I, TI and Termination II, TII). We provide evidence for the absence of AABW export from the Weddell Sea during the last and penultimate glacial maximum into the Atlantic sector of the SO. While the deglacial southward migration of the SO fronts is clearly resolvable in the Pb isotopic record, the Nd isotope data suggest that export of Weddell Sea-derived AABW resumed late during both glacial terminations, coinciding with the last major atmospheric $CO_2$ rise in the transition to the Holocene and the Eemian. Overall our data suggest that Weddell Sea AABW export can be reduced or absent during both colder and warmer climates than current.

## Results

**Sediment core sites**. We generated seawater Pb isotope records ($^{206}Pb/^{204}Pb$ and $^{208}Pb/^{204}Pb$) from a sediment core in the Atlantic sector of the SO (ODP Leg 177, Site 1094, 53.2°S, 5.1°E, water depth 2807 m; Fig. 1), and the corresponding bottom water Nd isotope records ($^{143}Nd/^{144}Nd$, expressed as $\varepsilon_{Nd}$) from authigenic Fe–Mn oxyhydroxides of two additional sediment cores: one located very close to ODP Site 1094 (PS1768-8, 53.6°S, 4.5°E, water depth 3299 m) and another located on the northern continental slope of the Antarctic Filchner–Rønne Ice Shelf (PS1599-3, 74.1°S, 27.7°W, water depth 2487 m; Fig. 1). The two northern sites are ideally located to sensitively record past changes in SO circulation due to their position in the mixing zone between Circumpolar Deep Water (CDW) and Weddell Sea Deep Water (WSDW) (Fig. 1)[18,21]. Specifically, at ODP Site 1094 and PS1768-8, covariation of Pb and Nd isotopic trends dominantly traces variations in the lower overturning cell, while Pb isotopic changes that have no resemblance in the Nd isotopic record reveal processes within the upper overturning cell. At the same time, the southern core from the East Antarctic margin enables us to monitor the $\varepsilon_{Nd}$ water mass signature of cold dense shelf water mixing with ambient water masses on the continental slope to the north of the Filchner–Rønne Ice Shelf[18].

**Pb isotopic evolution during Termination I**. Least radiogenic (lowest) Pb isotope compositions at ODP Site 1094 were recorded during the Last Glacial Maximum (Fig. 2b, c). Both $^{206}Pb/^{204}Pb$ and $^{208}Pb/^{204}Pb$ experienced a major change to significantly more radiogenic Pb isotopic compositions during the early deglaciation until the onset of the Antarctic Cold Reversal (~15 ka BP) during which the ambient $\varepsilon_{Nd}$ at Site PS1768-8 was invariant (Fig. 2g), suggesting changes in the upper SO-overturning cell only. The change in Pb isotope composition during this interval is very steady over more than 3 ka. It coincides with the first rise in atmospheric $pCO_2$ (Fig. 2a) and correlates with indicators of

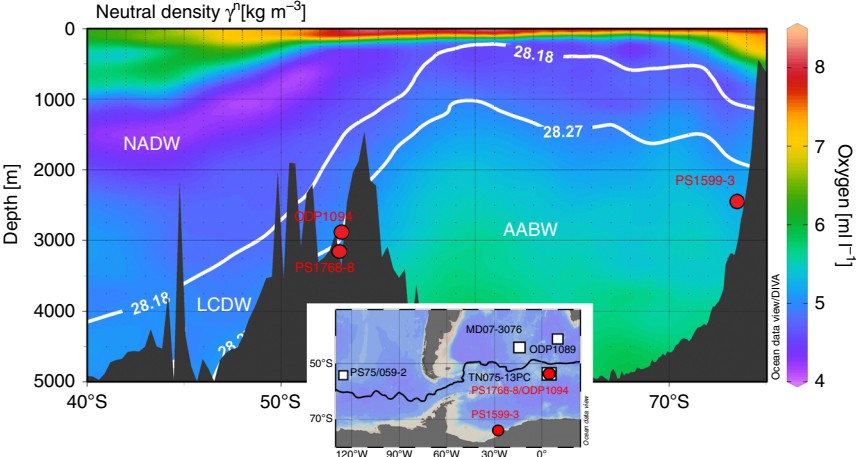

**Fig. 1 Studied sediment core sites.** Color-mapped oxygen concentrations, with neutral density contours ($\gamma^n$) overlain, as drawn from the 2013 World Ocean Atlas[69,70]. Sediment cores presented here (ODP Site 1094, PS1768-8 and PS1599-3) are marked with red circles. The locations of other cores mentioned in the text are marked as white squares in the inlet map. The isopycnal with a neutral density of 28.27 kg m$^{-3}$ defines the water mass boundary between AABW and LCDW, while the 28.18 kg m$^{-3}$ isopycnal defines the top of Antarctic Circumpolar Current bottom water as defined in Orsi et al.[18]. Figure created with Ocean Data View[71]. AABW: Antarctic Bottom Water; LCDW: Lower Circumpolar Deep Water; NADW: North Atlantic Deep Water.

increasing upwelling in the Atlantic sector of the SO (i.e., a decrease in deep SO radiocarbon reservoir ages[5] (Fig. 2d), and increasing opal flux[2] (Fig. 2e)). In $^{206}Pb/^{204}Pb$–$^{208}Pb/^{206}Pb$ isotope space (Fig. 3), the isotopic signature during the glacial interval of our records resembles most closely hydrogenic Pb isotope compositions seen at the Drake Passage and the SW Atlantic, trending toward eastern Weddell Sea compositions throughout the deglaciation. We can exclude a dust, anthropogenic or IRD control on the observed Pb isotopic changes (see Supplementary Figs. 3, 5, and 6). The steadiness of the Pb isotopic shift (Fig. 2b, c) during the early deglaciation argues for a gradual oceanic process on millennial timescales, and we interpret these changes as corresponding to a gradual southward displacement of major oceanic fronts in the SO during the first half of Termination I until the Antarctic Cold Reversal (ACR)[10] (Fig. 2b, c).

**$\varepsilon_{Nd}$ variation during Termination I.** In context of the early deglacial Pb isotopic evolution of the SO water column, a key observation in our records can be made in the bottom water Nd isotopic record of Site PS1768-8. The $\varepsilon_{Nd}$ seen during the late LGM and most parts of the deglaciation until 12.7 ka BP is on the order of ∼−4 (Fig. 2g). Such a composition is uncommon for the Atlantic sector of the SO that features a modern composition of −8.5 ± 0.39 $\varepsilon_{Nd}$[32], implying entirely different water mass sourcing and circulation regime during glacial climate compared with the present-day situation. The $\varepsilon_{Nd}$ remained very radiogenic until late into the deglaciation, when it changed to compositions close to modern Lower Circumpolar Deep Water (LCDW) $\varepsilon_{Nd}$ within about 2000 years. The clearly defined change in $\varepsilon_{Nd}$ was associated with a second smaller excursion toward more radiogenic Pb isotopic compositions (Fig. 2b, c), an increase in East Antarctic ice core EDML $\delta^{18}O^{33}$ (Fig. 2j), and the second rise in atmospheric $pCO_2$ (Fig. 2a), and coincided with increasing Ross Sea AABW export in the Pacific sector of the SO[34] (Fig. 2h).

## Discussion

The glacial deep-water Pb isotopic composition (25 ka–18 ka) agrees with water mass sourcing from the Drake Passage and the SW Atlantic (Fig. 3). The contemporaneous very radiogenic $\varepsilon_{Nd}$ in PS1768-8 was clearly different from Nd isotope compositions recorded within the Weddell Sea at Site PS1599-3 at the end of the LGM (19 ka–18 ka) (Fig. 2g). Active export of AABW during

the late LGM from the Weddell Sea should transfer Weddell Sea Pb and Nd isotope signatures to our two northern SO sites, yet such a signal is not observed. Our results therefore suggest the absence of Weddell Sea-derived AABW export to the Atlantic sector of the SO during the LGM (illustrated in Fig. 4a). As indicated above, onset or invigoration of AABW export should result in simultaneous changes in our Pb and Nd isotope records. In contrast, the $\varepsilon_{Nd}$ in PS1768-8 remained constant and radiogenic until 13 ka, while only Pb isotope composition attained eastern Weddell Sea signatures already before the onset of the ACR (Fig. 2b, c). Since the deep marine Pb isotope signal is influenced by changes in the entire water column, but the deep Nd isotope signature only responds to transitions in bottom water sourcing, we suggest that the decoupled Pb isotope evolution between 18 and 13 ka was merely controlled by circulation changes in the upper water column without significant Weddell Sea-derived AABW export. Given the strikingly steady increase in $^{206}Pb/^{204}Pb$ and $^{208}Pb/^{204}Pb$ during the early deglacial until ∼14.7 ka at ODP Site 1094, its covariation with atmospheric $pCO_2$ (Fig. 2a), and correlation with other proxies indicative for increasing upwelling during this interval (Fig. 2d, e), we regard the steady Pb isotopic shift as tracing the southward displacement of the SO fronts, resulting in increasing admixtures of Weddell Sea-sourced near-surface waters (illustrated in Fig. 4b). This interpretation is further supported by water column $\delta^{18}O$ and $\delta^{13}C$ reconstructions of the Atlantic sector of the SO that recorded water column isotopic changes only in water depths shallower than ∼2.5 km during the early deglacial[35]. We also note that the Pb isotope records define a plateau during the early ACR, which corresponds to the timing of Meltwater Pulse 1A (MWP1a). A high-resolution iceberg-rafted debris record from the Scotia Sea identified this time interval as the most prominent ice-rafting event from the Weddell Sea during TI[36]. A recent modeling study furthermore suggested increased meltwater rates in the Weddell Sea sector of the Southern Ocean after Heinrich Stadial 1, which may have temporarily affected overturning strength in the SO[37]. The Pb isotope evolution at ODP Site 1094 throughout the ACR agrees with such a scenario.

The Nd isotope record from Site PS1768-8 in turn contains key information on abyssal overturning dynamics in the lower SO circulation cell. Generally, Nd isotopic changes in South Atlantic settings are usually more gradual[38], while $\varepsilon_{Nd}$ changes, such as

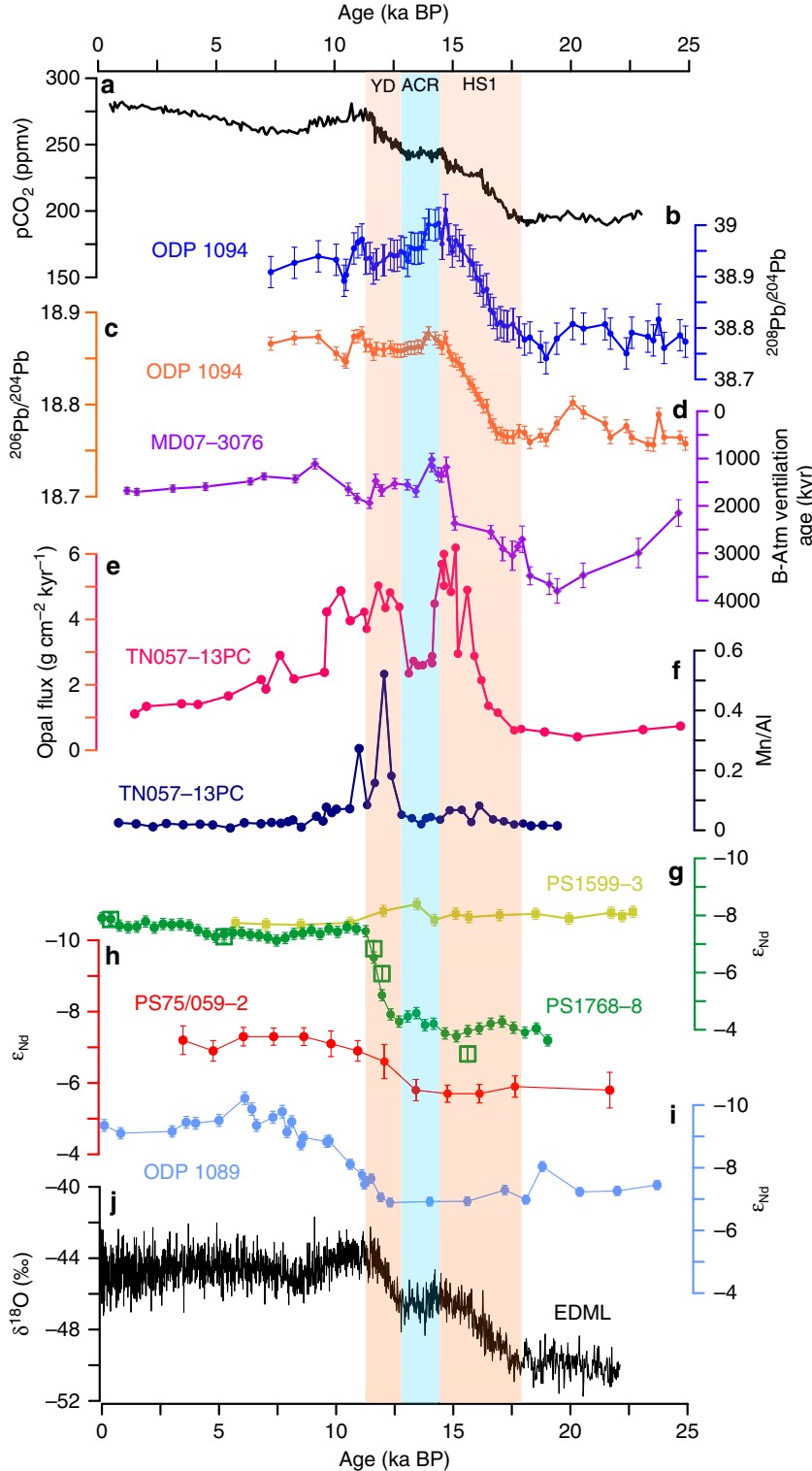

**Fig. 2 Paleoclimatic reconstructions over the past 25,000 years. a** Atmospheric $pCO_2$ concentrations from Antarctic Dome C ice core[6] and the West Antarctic Ice Sheet Divide ice core (WDC)[11]. **b**, **c** Authigenic Fe–Mn oxyhydroxide-based $^{206}Pb/^{204}Pb$ and $^{208}Pb/^{204}Pb$ records from ODP Site 1094. **d** Benthic–atmospheric (B–Atm) ventilation age[5]. **e** $^{230}Th$-normalized biogenic opal flux[2]. **f** TN057-13PC sedimentary Mn/Al[4]. **g** $\varepsilon_{Nd}$ records extracted from bulk sediment from PS1599-3 (yellow–green circles) and PS1768-8 (darker green circles) and opal from PS1768-8 (open green squares). **h** Fish teeth and planktic foraminifera $\varepsilon_{Nd}$ of South Pacific core PS75/059-2[34]. **i** $\varepsilon_{Nd}$ records extracted from ODP Site 1089[39]. **j** EDML ice core $\delta^{18}O$[33]. Error bars correspond to the $2\sigma$ external error of the $^{206}Pb/^{204}Pb$, $^{208}Pb/^{204}Pb$, and $\varepsilon_{Nd}$ measurements. The orange and blue boxes indicate periods of strong and diminished deglacial $CO_2$ rise. YD: Younger Dryas; ACR: Antarctic Cold Reversal; HS1: Heinrich Stadial 1.

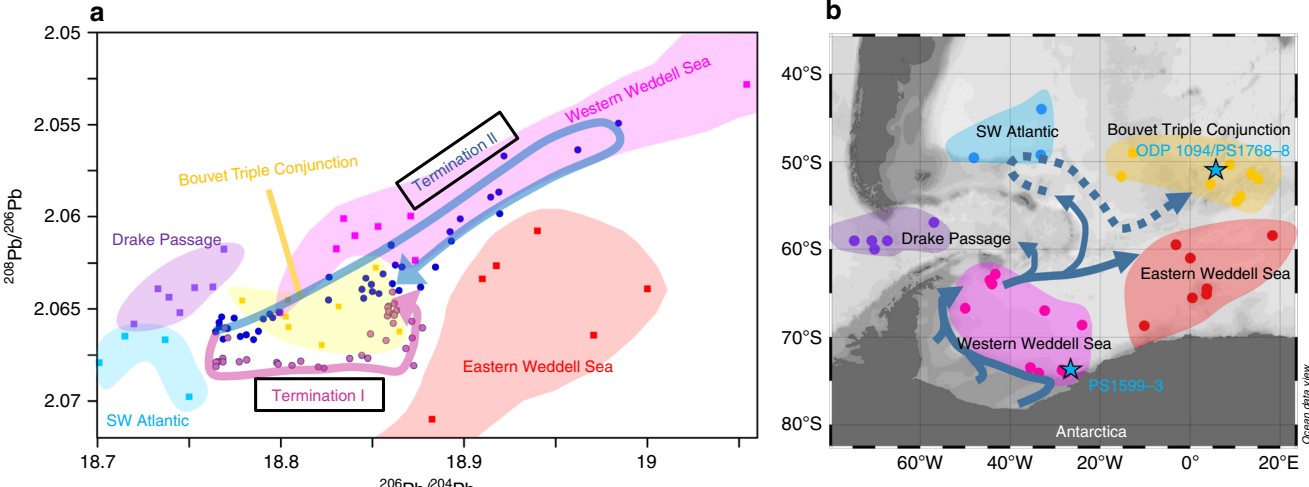

**Fig. 3 $^{206}Pb/^{204}Pb$–$^{208}Pb/^{206}Pb$ trends during Terminations I and II. a** Authigenic Fe–Mn oxyhydroxide-based Pb isotopic evolution observed at Site 1094 compared with Pb isotopic signatures in surface Fe–Mn nodules[72] and compositions extracted from core-top sediment samples in different parts of the Atlantic sector of the Southern Ocean shown in (**b**) (see the section "Methods"). The regionally characteristic Pb isotopic signature for various regions is marked by different colors. The arrows delineate the Pb isotopic evolution across the last two glacial–interglacial transitions (Termination I arrow covers 18 ka−12 ka BP and Termination II arrow covers 142 ka−125 ka BP). **b** Locations of Fe–Mn nodules and core-top sediment samples (Supplementary Data 2). The blue arrows in (**b**) indicate the modern AABW circulation and export path. The dashed blue arrow traces the LCDW flow path after entrainment of Weddell Sea AABW[73]. Panel (**b**) created with Ocean Data View[71]. AABW: Antarctic Bottom Water; LCDW: Lower Circumpolar Deep Water.

those seen at ~13 ka here, are commonly observed in the deglacial North Atlantic[39]. The rapidity and magnitude of change in $\varepsilon_{Nd}$ together with synchronous $^{206}Pb/^{204}Pb$ and $^{208}Pb/^{204}Pb$ excursions hence argues for a local bottom water mass sourcing switch. While $\delta^{18}O$ and $\delta^{13}C$ water column properties below ~2.5-km water depth in the Atlantic sector of the SO did not change substantially before the late deglacial, a clear shift toward Holocene values was reported during this time window[35]. Given the water mass provenance information contained in the Nd and Pb isotope proxies, we can identify the transition to typical Weddell Sea $\varepsilon_{Nd}$ signatures that covaried with these deep ocean $\delta^{18}O$ and $\delta^{13}C$ changes as recording the onset of Weddell Sea Deep Water export into northern reaches of the SO at ~13 ka BP (illustrated in Fig. 4c). ODP Site 1089 located to the NE of Site PS1768-8 recorded simultaneous changes in $\varepsilon_{Nd}$ alongside our site (Fig. 2i). Yet at Site 1089 this switch is more gradual and offset to less radiogenic $\varepsilon_{Nd}$, reflecting its position further away from the exit route of Weddell Sea AABW, and also being located in the export path of NADW into the SO[38,39]. In contrast to Site PS1768-8, the southernmost $\varepsilon_{Nd}$ record from Site PS1599-3 (Fig. 2g) situated on the East Antarctic continental rise that continuously monitored Weddell Sea compositions is strikingly invariant over Termination I and converged with northern Site PS1768-8 at this time. Last but not least, a further study reported a sedimentary Mn/Al spike (Fig. 2f) in the nearby core TN057-13PC of PS1768-8[4], interpreted as being driven by an increased ventilation of AABW during this interval.

The absence of Weddell Sea AABW export into the northern reaches of the SO outside the Weddell Sea during glacial climates has not been reported to date. In prior studies a sluggish yet operational glacial AABW water mass presence in the SO was invoked in glacial circulation scenarios[10,40–42], although no study to date could localize the origin of this deep glacial SO water mass. We tested the integrity of our presented Nd isotope signal by extracting the pure opal fraction from the same sediments without adhering terrigenous material and gently extracted the authigenic signature from these. The $\varepsilon_{Nd}$ extracted from these opal samples is matching the bulk sediment leach results (Fig. 2g). The authigenic signal is also different from detrital $\varepsilon_{Nd}$ signatures

in the Holocene and late deglacial section of this core (Supplementary Fig. 1). Moreover, the $\varepsilon_{Nd}$ record appears very clear-cut, suggesting two water mass circulation regimes. We have hence good reason to suggest that the water mass change observed in our record at 13 ka BP is not controlled by analytical artifacts.

However, a second key information contained in our $\varepsilon_{Nd}$ record is the glacial deep-water composition with an $\varepsilon_{Nd}$ of ~−4. In the modern ocean such a composition is characteristic for North Pacific Deep Water (NPDW)[43]. Although NPDW contributes to LCDW today[44], no resolvable isotopic trace of this $\varepsilon_{Nd}$ signature can be found in the present-day Drake Passage water column[32] upstream from Site PS1768-8. Conversely, cold-water coral-based glacial and deglacial Upper Circumpolar Deep Water (UCDW) Nd isotope records presented from water depths between 700 and 1750 m suggest that $\varepsilon_{Nd}$ within UCDW at the Drake Passage did not become more radiogenic than an $\varepsilon_{Nd}$ of ~ −6 at any time during Termination I[45]. A previous record from the South Atlantic (Site MD07-3076, 44°4.46′S, 14°12.47′W, 3770-m depth) reported early deglacial $\varepsilon_{Nd}$ of ~−5.5[46]. Potentially, Nd can be released from marine sediments to deep marine bottom water under very sluggish circulation regimes[24,29]. In marine settings such as the North Pacific[28] or North Atlantic[29], where this process has been observed, the porewater signature (i.e., the $\varepsilon_{Nd}$ signal extracted here) is expected to be altered away from bottom water $\varepsilon_{Nd}$ toward other non-hydrogenetic sedimentary phases such as volcanic debris within the sediment[29]. If the early deglacial South Atlantic and Drake Passage water column signatures on the order of −5.5[46] to −6.0[45] were equally representative for our core sites in the Atlantic sector of the SO, then the radiogenic glacial and deglacial signatures on the order of ~−4 were affected by such porewater alteration, changing the authigenic $\varepsilon_{Nd}$ to some degree away from ambient bottom water compositions. Hence if Weddell Sea-derived AABW had been advected to Site PS1768-8 in a very sluggish deep-water circulation regime during earlier parts of the deglaciation, we could have missed its arrival. We cannot entirely exclude this possibility although we consider it very unlikely since onset of a stronger bottom water circulation regime should equally result in authigenic $\varepsilon_{Nd}$ that is closer to modern compositions. Whether the

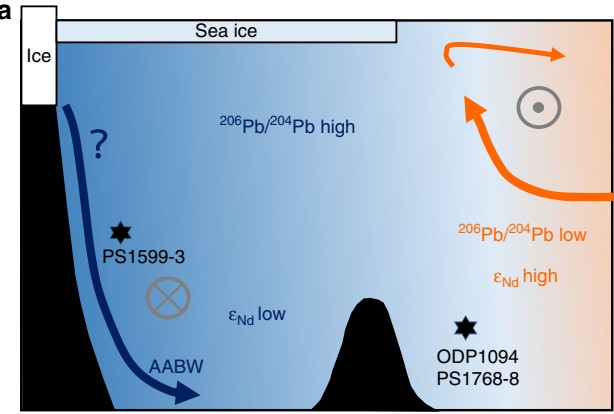

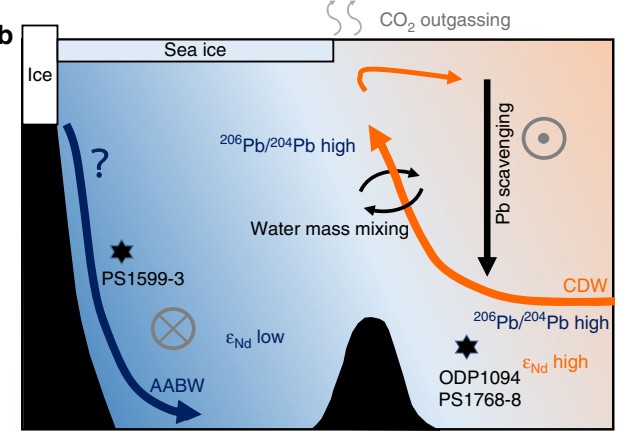

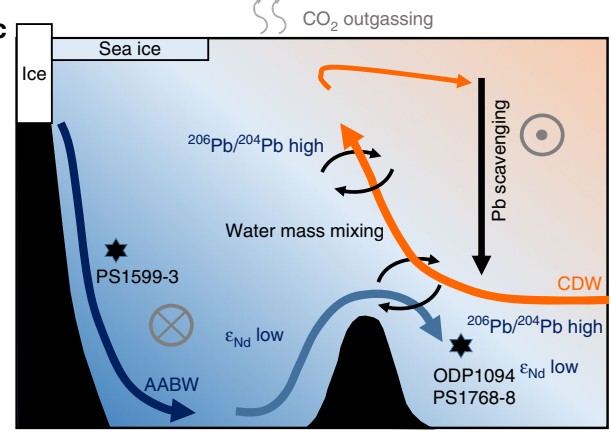

**Fig. 4 Hypothesized Southern Ocean circulation modes. a** LGM-overturning configuration with northward-shifted Southern Ocean circulation cell and absence of Weddell Sea-derived AABW at ODP Site 1094/Site PS1768-8, resulting in unradiogenic (low) $^{206}Pb/^{204}Pb$ and local radiogenic (high) $\varepsilon_{Nd}$ signature. **b** Early deglacial overturning as experienced preceding the Antarctic Cold Reversal (~15 ka). Southward displacement of Southern Ocean overturning cell, leading to entrainment of a Weddell Sea-sourced radiogenic (high) Pb isotope signal. **c** Post-Younger Dryas (YD) (since ~11 ka) and Holocene overturning configuration. Reinvigoration of AABW export resulted in the convergence of $\varepsilon_{Nd}$ inside and outside of the Weddell Sea. Crosses mark predominant westward flow directions and the point symbol denotes eastward flow. LGM: Last Glacial Maximum; AABW: Antarctic Bottom Water; CDW: Circumpolar Deep Water.

deglacial deep-water $\varepsilon_{Nd}$ was indeed affected by porewater alteration processes still needs to be confirmed. Since the modern extracted $\varepsilon_{Nd}$ compositions at Site PS1768-8 agree with direct seawater data, such an alteration was limited to (de)glacial climate stages and significantly reduced SO overturning.

To test whether our observed changes are indeed controlled by Milankovitch cycles forcing we also investigated the Pb and Nd isotope evolution at the same sites during Termination II (Fig. 5). The temporal change in incoming solar radiation during TII was significantly larger than that seen during TI[47], leading to monotonic warming over ~8 ka[48], peak Eemian temperatures at least 2 °C warmer than present, and global sea level some 4–6 m higher than today[49]. Furthermore, the atmospheric pCO₂ rise and warming of Antarctica was likely largely completed when Northern Hemisphere ice sheets started to melt[48]. These different boundary conditions surrounding TII lead to resolvable difference in our Pb and Nd isotopic records yet equally show striking similarities.

First of all, the changes in $\varepsilon_{Nd}$ again followed changes seen in Pb isotopes with a delay of several thousand years. Analogously to Termination I, Pb isotope compositions become more radiogenic during the penultimate deglaciation, yet after a short radiogenic spike at 132 ka BP compositions quickly drop to intermediate values (Fig. 5k, l). In $^{206}Pb/^{204}Pb–^{208}Pb/^{206}Pb$ isotope space the deglacial changes follow a parallel trajectory to changes seen during TI, yet more closely resembling western Weddell Sea compositions (Fig. 3) as opposed to a trend toward eastern Weddell Sea compositions recorded during TI. The strong Pb isotopic similarities with western Weddell Sea compositions could be a consequence of increased melting of the West Antarctic Ice Sheet during TII[50], yet this observation needs further investigation. The $\varepsilon_{Nd}$ record shows similarly radiogenic compositions in the (de)glacial section preceding 133 ka BP, followed by a less sharp switch to least radiogenic compositions coinciding with heaviest $\delta D$ values seen in the Vostok ice core, followed by rather intermediate $\varepsilon_{Nd}$ during MIS 5e.

The Eemian interglacial was warmer than the Holocene and the Antarctic Ice Sheet likely smaller[51]. One study investigating the authigenic uranium concentration at ODP Site 1094, including opal flux and Th$_{xs}$ measurements, reported an apparent overturning stagnation event in the SO in the middle of the last interglacial[52] as a consequence of increased melting of Antarctic continental ice. Both our Pb and Nd isotopic compositions follow a well-defined perturbation alongside the authigenic uranium excursion (Fig. 5c, j), suggesting reduced Weddell Sea-derived AABW export within the Eemian. Our Eemian Nd isotope record never reaches a classical Holocene SO signature, which should be on the order of ~−8, and is remarkably similar to the early deglacial Drake Passage cold-water coral record in UCDW[45] during the reported interval. Whether this intermediate $\varepsilon_{Nd}$ reflects the absence of Weddell Sea-derived AABW or is rather controlled by changes in SO overturning outside the Weddell Sea needs further research. Analogously to TI, however, the very radiogenic $\varepsilon_{Nd}$ signature and unradiogenic Pb isotopic compositions observed preceding TII are observable and argue for the absence of Weddell Sea AABW export during the penultimate glacial maximum.

The lack of Weddell Sea AABW export into the Atlantic sector of the SO during the last two glacial maxima does not necessarily imply that this water mass was not formed, rather that it was too dense to be exported to our northern SO core sites. Such a scenario is not unlikely since a reduced Weddell Gyre circulation could lead to reduced diapycnal mixing within the basin. The Weddell Sea is bound by the South Scotia Ridge and the Bouvet Triple Junction to the North. While Weddell Sea Bottom Water is too dense to leave the Weddell Sea today, the admixture of

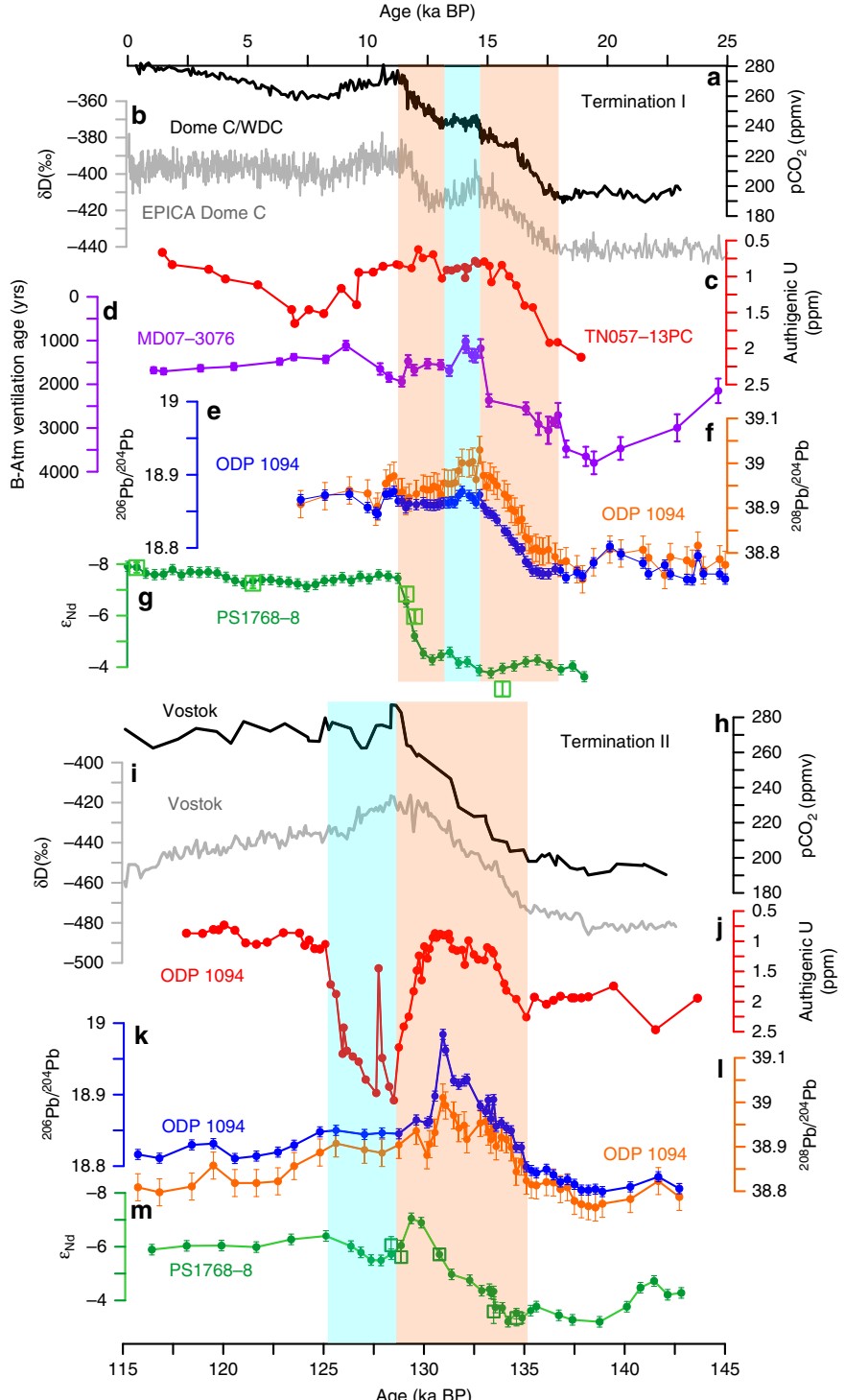

**Fig. 5 Reconstructions covering Terminations I and II. a** Atmospheric $pCO_2$ concentrations from Antarctic Dome C ice core[6] and the West Antarctic Ice Sheet Divide ice core (WDC)[11]. **b** δD[74] from Dome C and EPICA Dome C ice cores. **c** Termination I sedimentary authigenic U concentrations at Site TN057-13PC[4]. **d** Benthic–atmospheric (B–Atm) ventilation ages[5]. **e**, **f** $^{206}Pb/^{204}Pb$ and $^{208}Pb/^{204}Pb$ records in Termination I from ODP Site 1094. **g** Termination I $\varepsilon_{Nd}$ records extracted from bulk sediment (closed green circles) and from opal (open green square) in core PS1768-8. **h**, **i** Atmospheric $pCO_2$[58] and δD[75] from Vostok ice core. **j** Termination II sedimentary authigenic U concentrations at ODP Site 1094[52]. **k**, **l** $^{206}Pb/^{204}Pb$ and $^{208}Pb/^{204}Pb$ records in Termination II from ODP Site 1094. **m** Termination II $\varepsilon_{Nd}$ records extracted from bulk sediment (closed green circles) and from opal (open green square) in core PS1768-8. Error bars correspond to the 2σ external error of the $^{206}Pb/^{204}Pb$, $^{208}Pb/^{204}Pb$, and $\varepsilon_{Nd}$ measurements. The orange and blue boxes higlight periods of strong and diminished deglacial atmospheric $pCO_2$ rise.

modified CDW within the Weddell Gyre is adding the required buoyancy to enable Weddell Sea AABW export[53]. Reduced Weddell Gyre circulation and associated lowered inflow of modified CDW, as well as extended sea ice coverage, may provide the boundary conditions required for the cessation of AABW export from the Weddell Sea. Given the bathymetry of the Weddell–Enderby abyssal plane we cannot exclude that a glacial equivalent of Weddell Sea AABW escaped the basin in deepest parts toward the east yet the northern export routes were closed. The clearly resolvable increase in atmospheric $pCO_2$ late during TI and TII (Fig. 5) in turn illustrates how reinvigoration of Weddell Sea AABW ventilation contributed to partition carbon out of the ocean interior. As such, the resumption of AABW may have been instrumental in driving the deglacial sequence of events to completion.

## Methods

**Core positions.** Ocean Drilling Program (ODP, Leg 177) Site 1094 (53.2°S, 5.1°E, water depth 2807 m) and RV Polarstern sediment core PS1768-8 (53.6°S, 4.5°E, water depth 3299 m) were retrieved from the Atlantic sector of the Southern Ocean, south of the modern Antarctic Polar Front (Fig. 1). Core PS1599-3 (74.1°S, 27.7°W, water depth 2487 m) was retrieved from the continental slope north of the Filchner–Rønne Ice Shelf (Fig. 1). Additional sediment core tops, collected from the sediment core repository of the Alfred Wegener Institute in Bremerhaven (Germany), used for seawater Pb isotopic characterization in Fig. 3 are listed in Supplementary Data 2.

**Age model.** For Termination I, existing age models of PS1599-3[54], PS1768-8[55], and Site ODP 1094[56] are directly taken from previous studies. For Termination II, the age model for Site ODP 1094 and PS1768-8 was refined via alignment of planktonic foraminiferal $\delta^{18}O$[56,57] with Antarctic ice $\delta D$ from Vostok[58] assuming an in-phase relationship (Supplementary Fig. 2) as described in Jaccard et al.[59].

**Pb isotope measurements.** The seawater Pb isotope signal was extracted from the authigenic Fe–Mn oxyhydroxide fraction of bulk sediment from ODP Site 1094 and 20 additional core-top bulk sediment samples following the leaching procedure refined from previous studies[30,60]. About 0.3 g of sediment samples were weighed and homogenized for each measurement. The authigenic Fe–Mn oxyhydroxide fraction in the sediment was extracted by vortexing the bulk sediment in a diluted reductive solution (0.005 M hydroxylamine hydrochloride/1.5% acetic acid/0.003 M Na-EDTA solution buffered to pH 4 with NaOH) on a vortex mixer not longer than 10 s. The centrifuged trace metal solution was extracted and the Pb cuts purified by ion chromatography using AG1-X8 resin[61]. Our Pb isotope measurements were performed on a Thermo Scientific Neptune Plus MC-ICP-MS at GEOMAR, Kiel. Mass bias correction during Pb isotope measurements was done externally using the Tl-doping technique[62,63] spiked by a NIST997 Tl standard solution (Pb:Tl = ~4:1). Given that Tl and Pb fractionate slightly differently during ionization[64], $^{205}Tl/^{203}Tl$ were determined on a session-by-session basis so that NBS981 Pb isotope compositions matched published compositions[63,65] (Supplementary Data 1). Total procedural blanks were below 50 pg ($n = 18$) and are hence negligible. The reproducibility of the secondary standard USGS NOD-A-1 reproduced Pb isotope compositions with a precision of 0.007 for $^{206}Pb/^{204}Pb$ and 0.030 for $^{208}Pb/^{204}Pb$ ($n = 58$).

**Nd isotope measurements ($\varepsilon_{Nd}$).** The seawater Nd isotope signal was extracted from the Fe–Mn oxyhydroxide fraction of bulk sediment of PS1599-3 and PS1768-8 in an identical manner to our Pb leaching procedure. Eleven samples from core PS1768-8 were selected for the determination of the detrital sediment Nd isotope composition after preceding removal of the authigenic ferromanganese oxide phase[30]. The Rare Earth Elements (REE) were separated from the leachate by cation exchange chromatography using 50W-X8 resin followed by separation of Nd from the other REE using LN-Spec resin[66,67].

Nd isotope measurements were also performed on a Thermo Scientific Neptune Plus MC-ICP-MS at GEOMAR, Kiel. Instrumental mass fractionation was corrected by normalizing the measured ratio of $^{143}Nd/^{144}Nd$ to $^{146}Nd/^{144}Nd = 0.7219$ using the mass bias correction procedure of Vance and Thirlwall[64]. The measured Nd isotope ratios were normalized to the published $^{143}Nd/^{144}Nd$ value of 0.512115 for JNdi-1[68]. Total procedural blanks for Nd are below 30 pg and hence negligible ($n = 15$). Secondary standard solution NIST 3135a reproduced within 0.20 $\varepsilon_{Nd}$ throughout the course of this study ($n = 32$) (Supplementary Data 3).

Our $\varepsilon_{Nd}$ signal leached from surface sediment in core PS1768-8 is matching the actual seawater $\varepsilon_{Nd}$ signal[32]. Given the lack of foraminifera or fish teeth for comparison with the leachate Nd isotope data in the sediment, past seawater $\varepsilon_{Nd}$ was also extracted from the Fe–Mn oxyhydroxide fraction in nine separated biogenic opal samples picked from core PS1768-8 to confirm the validity of the leaching method. The fraction > 63 mm was dried and about 0.1 g of biogenic opal

was picked. The authigenic Fe–Mn oxyhydroxide signal contained in the opal fraction was extracted by the same method we used for bulk sediment leaching. Our results obtained from the leachates of the Fe–Mn oxyhydroxide fraction based on either bulk sediments or purified opal are in agreement and we can also exclude IRD control on the observed $\varepsilon_{Nd}$ evolution (see Supplementary Discussion and Supplementary Figs. 3 and 4). Therefore, the $\varepsilon_{Nd}$ evolution trend across the two terminations was reliably capturing a porewater signature[29].

## Data availability

Metadata and other results from the sediment cores are available within the Pangaea database (https://doi.org/10.1594/PANGAEA.909084).

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

## Acknowledgements

Sediment material for this study was obtained from the AWI core repository in Bremerhaven and the ODP/IODP core repository Bremen. We thank T. Goepfert, A. Kolevica, and M. Seebeck for technical support. This paper benefited from discussions with S. Jaccard, M. Frank, and J. Lippold. H. Huang acknowledges the China Scholarship Council (CSC) for providing financial support to his overseas study.

## Author contributions

M.G. conceived the study and initialized this project in cooperation with G.K. M.G., H.H. and G.K. selected the core sites and provided geochemical data. H.H. carried out the analytical work as well as the Pb and Nd isotopic analyses with guidance from M.G. and A.E. H.H. and M.G. wrote the paper. All coauthors contributed to the final version.

## Competing interests

The authors declare no competing interests.
