## [Peer Review File · Nature Communications]

Reviewers' comments:

Reviewer #1 (Remarks to the Author):

Huang et al. in the manuscript entitled 'No Weddell Sea Antarctic Bottom Water export during the Last and Penultimate Glacial Maximum' investigated three sediment cores from the Atlantic sector of the Southern Ocean. It has been long discussed in the Paleoceanography community that the (peak) glacial deep ocean (particularly the Atlantic) has been primarily flushed by AABW, however, there is no direct evidence as to the source of this deep water. Huang et al. in this contribution have addressed this controversy head on and in doing so they have studied a region that is close to the potential origin of this deep water mass. In this study, they report Fe-Mn oxyhydroxide derived Pb and Nd isotopic record to report that Weddell Sea was not an active site of Antarctic Bottom Water export during the last two peak glacial times. While this is an important problem and should be investigated and is of immense relevance, the conclusions and interpretations put forward in this manuscript are far from convincing. I am fully supportive of this study, however, this manuscript needs major revision and perhaps additional data before this can be deemed suitable for publication.

Detailed Comments:

The overall organization of content and inclusion of relevant figures can improve the quality of the manuscript. The authors talk about three cores that they have used in their study and then they have plotted several published records. However, there is no map in the main text and had to look into the supplementary information. So, it will be very helpful if the authors could include a map that shows the cores they have studied (in one color or symbol) and the published cores used in their figures and text (in another color or symbol). It will help the readers to related to the data in hand. Moreover, one ODV depth vs. salinity/neutral density figure showing the current position of the three cores would be useful to understand the modern position of the cores with respect to the water mass boundaries. As for the organization of the content, I will leave this to the authors, as there are more than one co-authors who have written numerous compelling papers and are well known for their excellent science.

The Pb isotope data is intriguing as it shows variations that are aligned with $\delta^{18}\text{O}$ changes observed in Antarctic ice core. However, I am not sure what did the authors conclude as to the reason for this change. In line 97-100, it is stated that Pb isotopic shift during the last deglaciation is due to oceanic processes. The authors mentioned of a possibility of southward displacement of front but it is not clear how does that relate to their observation? Do they indicate enhanced AABW export, if yes why would the Pb isotopes behave the way it behaved? A short description of how these Pb isotopic data have been interpreted would be beneficial to maintain the flow of the manuscript. I am curious why the authors did not report any Pb isotope data for core PS 1599-3. Would it not be really beneficial to have Pb isotope record for that core in order to understand AABW export to more northerly cores along the route of AABW export?

A major mismatch between Pb and Nd isotopic data presented in this manuscript is that the Nd isotopic record does not show the same pattern of change as the Pb isotopes do (for the last 30 kyr). In Line 113-115 the authors compared and contrasted their downcore ϵ_{Nd} record with that of other commonly observed South Atlantic record. In my opinion, this is an important section, which needs a bit more explanation. Such as, if they are observing patterns that are not normal to the South Atlantic (or to that region) then what is different and how can the authors account for these differences. Is there a pattern as to how sedimentary core north of certain latitude shows a particular deglacial behavior compared to the ones, which are located at higher latitudes?

The authors discussed the integrity of their down core Nd isotope data. I am very much in favor of the approach of using pure opal to generate Nd isotope record, it might be the only way if high-resolution records are to be generated for high latitude Southern Ocean. Being said that, the logic presented in the manuscript to establish that their records are indeed representative of bottom water is a bit circular. In the supplementary section Fe-Mn oxyhydroxide derived ϵ_{Nd} isotope from core PS 1768 are presented along with a few opal-derived data. While the opal derived ϵ_{Nd} numbers matched with downcore data, it is necessary to have a pure opal derived value for the core top. Should the core top Fe-Mn oxyhydroxide, core top pure opal, and ambient bottom water

eNd values match, then it would establish beyond doubt that the generated data is not an artifact of their method. Another approach could have been to compare the carbonate content of the core and Fe-Mn oxyhydroxide derived eNd record. If there is high correlation between how the carbonate content changed in the core and how the downcore eNd changed, then it would be difficult to argue that Fe-Mn oxyhydroxide derived data are pure bottom water.

In line 137, the authors claimed 'The authigenic signal is also different from and does not co-vary with the detrital eNd signatures...'. I agree that the authigenic values are different but it is unclear if they co-vary. The detrital record is low resolution and the overall pattern of change in the detrital record has a resemblance to the Fe-Mn oxyhydroxide record for the last ~20 kyr.

The downcore eNd record in core PS1768-8 has values around -4, which the authors have interpreted to be due to sedimentary boundary release processes. They further claimed that it is only prevalent during times of sluggish circulation i.e during glacial and early deglacial times. However, if this is true, then is it not possible that any changes in the downcore Nd record that could have been due to AABW export during the glacial and early deglacial times might have been overprinted by sedimentary processes?

It is also not clear to me how the authors arrived at the conclusion of no AABW export during the glacial. The glacial part of the record in PS 1768-8 is really short (and does not even cover LGM) and moreover, it is claimed to be overprinted by sedimentary processes. Is this conclusion based on Pb isotopes? If so then there needs to be a clear explanation. A water mass mixing exercise to calculate the relative proportion of AABW and other relevant water masses might be useful.

I am supportive of this manuscript with a major revision.

Reviewer #2 (Remarks to the Author):

This paper uses lead isotopes and eNd (epsilon Nd) to document the changing contribution of Weddell Sea bottom water to the South Atlantic sector of the Southern Ocean over the last two glacial terminations. In this regard it is similar to Basak et al. (Nature, 359, 900-904, 2018) who used changes in eNd to look at the contribution of Ross Sea bottom water to the Pacific sector of the Southern Ocean. The problem is that the eNd story in the present paper is not nearly as clear cut as it is in the Pacific. The Pb isotope story, on the other hand, seems to be very informative. Hence, my recommendation is that the authors jettison the eNd data in the paper and focus instead on the Pb isotope data.

General readers, like myself, could use some more background on the Pb isotopes in the South Atlantic. The authors obviously went to some trouble to compile Pb isotope compositions from dozens of core tops across a large area (Fig. 2) and they found a wide range of compositions that reflect on the downcore compositions at ODP site 1094. The comparison shows that the deep water at site 1094 seems to have been dominated by relatively unradiogenic water with origins in the Pacific or South Atlantic during the last two glacial periods. But then more radiogenic water began to be swept out of the Weddell Sea starting about 18,000 and 135,000 years ago. The implication is that the more radiogenic compositions were confined to the Weddell Sea during the prior glacial periods because there was no production of Weddell Sea bottom water to carry these compositions out to the larger area beyond. This is the most instructive finding in the paper. But how robust is it? Why are

the Pb isotope compositions so much more radiogenic in the eastern and western Weddell Sea in relation to Drake Passage and the SE Atlantic?

Reviewer #3 (Remarks to the Author):

Review of Huang et al.

Huang et al. investigate changes in Weddell Sea derived Antarctic Bottom water across the last and penultimate deglaciation, and the Eemian. For this they present Pb isotope records from Southern Ocean core ODP1094, as well as Nd isotope records from nearby core PS1768-8 and Weddell Sea core PS1599-3. ODP1094 and PS1768-8 cores cover the last deglaciation (~20-7ka) as well as the penultimate deglaciation and the Eemian (~143-115 ka), while core PS1599-3 only covers the last deglaciation.

Given the importance of AABW for the global carbon cycle, gaining a better understanding on sources and temporal changes in AABW across both deglaciations and the Eemian is crucial. I think this study does make a step in this direction by presenting original and relatively high-resolution records from the Southern Ocean. So, in principle I would support the study for publication in Nature Communications. However, below are some comments that would first need to be addressed.

1) Interpretation of Pb isotopes

The eps(Nd) record of PS1599-3 in the Weddell Sea show little to no change across the deglaciation and the eps(Nd) record from PS1788-8 reaches the value of PS1599 rather rapidly. The authors interpret this as reflecting the dominance of Weddell Sea waters, which formation could have been quite influenced by the ice-sheet retreat over the Weddell Sea.

The Pb isotopes records from ODP1094 are quite interesting as they could reflect less local changes. I should flag that this is the first time I come across Pb isotopes records, so I have no knowledge on this proxy. Given the authors' explanation of the proxy, the data shown in Figure 2 and the records, I can be convinced that the increase in 208/204Pb and 206/204Pb could be due to an increased proportion of Western Weddell Sea waters.

However, the authors write L.98-99, that the Pb changes are due to a southward displacement of the front as seen in Xiao et al. Here, we are not talking about fronts, but about the location of formation, pathway (and strength?) of "southern-sourced deep waters".

In addition, it would be good to know the Pb isotopes signature of NADW (I could not find it). This is particularly important since ODP1094 is not that deep. Maybe some additional information on Pb isotopes could be added to the supplementary information.

2) Scope of the study

The study presents records covering both deglaciation and the Eemian. One could argue that this could be split into 2 or 3 papers. I understand why the authors would want to show everything in one paper, but then the scope of the study becomes quite large and the details given in the different topics become quite small. Additional information and details on the LGM and the last deglaciation could be given if the study was to focus on that. It would make the study clearer. The penultimate deglaciation is only briefly mentioned with no background information at all. The Eemian is rushed through...

3) Introduction

The introduction would need to be more precise and associated references more appropriate.

- L.35-36: "pCO₂ increased in 2 main intervals": please state which intervals you are referring to. Also, more recent higher resolution atmospheric CO₂ records have been published: e.g. Marcott et al. 2014 Nature.

- L. 37-38: I am not really sure "a consensus could be reached" that it was due to a "southward moving overturning cell". Changes in the southern hemispheric westerlies have been proposed (2,8), and they would have had an impact on the SO overturning cell and atmospheric CO₂ (see also Menviel et al., 2018), pointing to a strengthening of the SO overturning, but whether it was shifted southward is up to debate, and in fact this is where your study is quite important.

- L. 39: I don't think ref. 6 is appropriate here. Ref. 2 already points to a stronger upwelling during the deglaciation.

- L. 39-42: This paragraph is really confusing, particularly in the context on what was discussed above. To which deglacial pulse are you referring to? "The bipolar seesaw" already operated during Heinrich 1, and is a term that means nothing and everything. Why would you discuss AMOC reductions only for "the second pulse"? The strongest AMOC reduction during the deglaciation most likely occurred during Heinrich 1 (e.g. Ng. et al., 2018).

4) Weddell Sea waters and SO overturning

This comment refers to L. 129-132 in particular, but it also refers to a comment made above, and would push for a more precise use of terms throughout the manuscript.

The precursor of AABW today are dense shelf waters (DSW) formed on the Antarctic continental shelves by brine rejection. The main regions of formation are the Weddell and Ross Seas. While the authors are completely correct that i) the location of formation of DSW could have changed in the past, ii) there is no study particularly looking at how the sources changed in the past, several studies have already pointed out that due to the Antarctic ice-sheet advance in the Ross and Weddell Seas, AABW formation might have been quite different in the past (e.g. Paillard & Parrenin, 2004, EPSL, Menviel et al., 2017). In their discussion, the authors should take into account the Antarctic ice-sheet advance (and subsequent retreat) over the Weddell Sea (Golledge et al., 2014 present a potential evolution).

In addition, it is quite possible that the AABW transport changed significantly. While I agree that AABW transport at the LGM could have been reduced, I am not sure this view would fit as "commonly" (L. 131), and I am not sure that the references included in that sentence are all appropriate. Inclusion of Ref. 1 could be discussed. Ref. 2 nicely infers a stronger Southern Ocean upwelling during Heinrich 1, however there is no indication of AABW formation at the LGM in their study.

-L. 195-196: "too dense to be exported to our northern SO core site". I understand that under special circumstances (and in modern day), if the "water is too dense" it could be blocked by topographic features, however in your case I don't think there is any evidence for such a mechanism at play. The authors might want to consider here the ice-sheet advance over the Weddell Sea (Golledge et al., 2014, Bentley et al., 2014) at glacial maxima. (L. 202-205) Given the location of your cores, an eventual blockage on the northern route should not matter.

Figure 2: It would be easier for the reader if the y axis was 208/204 Pb, as shown for core ODP1094 in figures 1 and 4.

Figure 3: I understand that the authors want to include a diagram, but in its present form it is not possible to show this figure as several aspects are not substantiated.

Given the text and records, the authors are really looking at the Weddell Sea, therefore the figures should really reflect that this is the Weddell Sea deep waters that are represented here and not the "Southern Ocean".

Then several aspects of the figure could be discussed/argued: i) HS1, YD are not defined nor really discussed in the text, ii) where does the sea-ice edge comes from? I can understand that the LGM sea-ice edge is further than the c) one, but what about b)? iii) where does the changes in the CDW arrow come from? iv) where do the water mass mixing changes come from? v) whereas the study mostly reflect changes in the source of AABW it is referred in the same way in the 3 diagrams, the only difference being "AABW going over the hump in c". Given the position of the cores, I'm not sure what the "hump" refers to.

Finally, one could argue about the relative depth positions of the cores with respect to their real positions.

Minor points and typos:

- L. 76: Core PS1599 is at 27 degrees West (and not East)!

- CDW is not defined (and nor are UCDW and LCDW later in the text)

- L. 81-82: I am not sure what the authors define here as "upper and lower SO overturning cells", and not sure that "the records resolve changes in the upper and lower cells". I would agree with the "lower cell" as changes in AABW composition, but I can't see where the "upper one" comes in.

- L. 153: "where"

- L. 163: I am not sure the changes the authors are seeing are due to changes in insolation and the term "Milankovitch forcing" might not be appropriate here.

No Weddell Sea Antarctic Bottom Water export during the Last and Penultimate Glacial Maximum

by

Huang Huang, Marcus Gutjahr, Anton Eisenhauer and Gerhard Kuhn

Specific points raised by the reviewers:

Reply #1: We thank the three referees for their critical yet generally positive perception of our manuscript and incorporated the suggested changes as much as possible. When reading the reviewer's comments, we also realised that we need to better explain key features of our proxies, particularly since the authigenic Pb isotope proxy was hitherto not employed for such paleoceanographic applications in the Southern Ocean (SO). In fact, the observation that authigenic Pb isotopes trace the deglacial frontal movement in the Southern Ocean so neatly came somewhat unexpected to us and is another key outcome of our study that should initiate follow-up studies.

We hope that the current version of the manuscript is better to follow to convey the key implications of our records. We still included the Termination II record since we also consider these data an essential part of our current manuscript. Both the similarities and differences between the paleoceanographic changes during Terminations I and II (TI and TII) demonstrate that subtle differences in atmospheric boundary conditions (i.e. the temporal degree of change of incoming solar radiation during each Termination) can make a significant difference for SO overturning circulation, Antarctic ice sheet disintegration, and AABW export.

Please note that line numbers referred to below correspond to those of the revised version of the manuscript with no highlighted changes.

Reviewer #1

Huang et al. in the manuscript entitled 'No Weddell Sea Antarctic Bottom Water export during the Last and Penultimate Glacial Maximum' investigated three sediment cores from the Atlantic sector of the Southern Ocean. It has been long discussed in the Paleoceanography community that the (peak) glacial deep ocean (particularly the Atlantic) has been primarily flushed by AABW, however, there is no direct evidence as to the source of this deep water. Huang et al. in this contribution have addressed this controversy head on and in doing so they have studied a region that is close to the potential origin of this deep water mass. In this study, they report

Fe-Mn oxyhydroxide derived Pb and Nd isotopic record to report that Weddell Sea was not an active site of Antarctic Bottom Water export during the last two peak glacial times. While this is an important problem and should be investigated and is of immense relevance, the conclusions and interpretations put forward in this manuscript are far from convincing. I am fully supportive of this study, however, this manuscript needs major revision and perhaps additional data before this can be deemed suitable for publication.

Reply #2: We thank the reviewer for these positive critical comments, which we are happy to answer below.

Detailed Comments:

The overall organization of content and inclusion of relevant figures can improve the quality of the manuscript. The authors talk about three cores that they have used in their study and then they have plotted several published records. However, there is no map in the main text and had to look into the supplementary information. So, it will be very helpful if the authors could include a map that shows the cores they have studied (in one color or symbol) and the published cores used in their figures and text (in another color or symbol). It will help the readers to related to the data in hand. Moreover, one ODV depth vs. salinity/neutral density figure showing the current position of the three cores would be useful to understand the modern position of the cores with respect to the water mass boundaries. As for the organization of the content, I will leave this to the authors, as there are more than one co-authors who have written numerous compelling papers and are well known for their excellent science.

Reply #3: A map, showing the locations of the cores used in this paper, has been added as Fig. 1 in the main text. This map also reveals the hydrographic context of the cores studied in this paper in an ODV generated section of the Atlantic sector of the SO, as requested by the reviewer. The core sites map in the supplementary discussion is then deleted. We further have reorganised the manuscript in a manner as to allow sufficient space to introduce key details/facts.

The Pb isotope data is intriguing as it shows variations that are aligned with $\delta^{18}\text{O}$ changes observed in Antarctic ice core. However, I am not sure what did the authors conclude as to the reason for this change. In line 97-100, it is stated that Pb isotopic shift during the last deglaciation is due to oceanic processes. The authors mentioned of a possibility of southward displacement of front but it is not clear how does that relate to their observation? Do they indicate enhanced AABW export, if yes why would the Pb isotopes behave the way it behaved? A short description of how these Pb isotopic data have been interpreted would be beneficial to maintain the flow of the manuscript. I am curious why the authors did not report any Pb isotope data for core PS 1599-3. Would it not be really beneficial to have Pb isotope record for that core in order to understand AABW export to more northerly cores along the route of AABW export?

Reply #4: A key point we absolutely want to convey to readers is the information

stored in the authigenic Fe-Mn oxyhydroxide-derived Pb isotope composition, which we now better explain in Lines 69-82 in the change accepted version. Importantly, past seawater Pb isotopic compositions extracted from deep marine sediments reflect the average composition of the *entire water* column, while the Nd isotopic composition extracted from deep marine sediments is usually dominated by the *bottom water* signature.

The nature of the signal recorded in the authigenic Pb isotope composition can best be illustrated via revisiting Figs. 2 and 3 (as also outlined in the manuscript). In the modern SO overturning system in the Atlantic sector of Southern Ocean, a part of modern CDW upwelling water gains buoyance at surface and moves northwards while another part of the upwelled CDW is advected southward towards the Antarctic shelf ice edge and sinks as newly formed AABW. These two processes are part of the upper and lower Southern Ocean overturning circulation cells (Talley, 2013). In our records the Pb isotope compositions change from less radiogenic to more radiogenic compositions during the early deglaciation, following a compositional trend moving from Drake Passage / South Atlantic signatures towards compositions seen in coretop sediments within the Eastern Weddell Sea (see Fig. 3). We know, however, from the Nd isotope record that this signal was not advected to ODP Site 1094 within AABW, otherwise the Nd isotope composition would need to be much less radiogenic during the early deglaciation (an ϵ_{Nd} of ~ -8 as opposed to -4 ; see Fig. 2g). A surface water Pb isotope signal was recently reported to be effectively transferred through the water column in particulate form to abyssal depths (Wu et al., 2010), and a concentration profile of dissolved seawater Pb would reveal a “scavenged elemental profile” with lower dissolved concentrations at depth than at shallow water levels. The only viable process to generate the early deglacial continuous change to more radiogenic compositions at ODP Site 1094, changing in concert with atmospheric pCO_2 , is hence a successive southward displacement of the SO oceanic fronts. Several publications identified the southward displacement of the ACC (and hence the SO oceanic fronts) during the early deglaciation as a key driver of rising atmospheric pCO_2 during this period (Anderson et al., 2009; Shakun et al., 2012; Toggweiler, 1999). The mechanism that brings Weddell Sea-derived Pb into the northern reaches of the SO must be the southward movement of the polar front, which diverts increasing amounts of shallow Weddell Sea water north, while overturning dynamics in the deeper SO cell (Ferrari et al., 2014) remained much reduced as indicated by our exotic Nd isotope signal until ~ 13 ka. At closer inspection the onset of Weddell Sea-derived AABW during the late deglacial leads to a further (smaller) excursion in Pb isotope compositions (Fig. 2c,c,g) yet this change is relatively muted compared with the early deglacial trends since ODP Site 1094 was already dominated by Pb supply from the Weddell Sea advected to the core site within the upper water column.

The reason why we did not generate an authigenic Pb isotope record at Site PS1599-3 is because Pb isotopic signal in PS1599-3 is dominated by the local

continental weathering signal and not reflecting an open Weddell Sea seawater signature. As shown in the supplemental table, the Pb isotopic signature of coretop sediments at Site PS1599-3 is very radiogenic with $^{206}\text{Pb}/^{204}\text{Pb}=19.12$, which is offsets from surrounding coretop authigenic Fe-Mn oxyhydroxide compositions and Fe-Mn nodule samples. Since Site PS1599-3 is very close to the Antarctic ice shield, its more radiogenic Pb isotopic signal is dominated by sub-glacially sourced local Pb overprinting advected signatures. Instead, we mapped coretop sediments in the Atlantic sector of the Southern Ocean and particular the Weddell Sea margin for its modern authigenic Pb isotopic variability (Fig. 3) to trace regional water mass sources whilst eliminating the continental weathering signal interference from individual core sites.

A major mismatch between Pb and Nd isotopic data presented in this manuscript is that the Nd isotopic record does not show the same pattern of change as the Pb isotopes do (for the last 30 kyr). In Line 113-115 the authors compared and contrasted their downcore ϵ_{Nd} record with that of other commonly observed South Atlantic record. In my opinion, this is an important section, which needs a bit more explanation. Such as, if they are observing patterns that are not normal to the South Atlantic (or to that region) then what is different and how can the authors account for these differences. Is there a pattern as to how sedimentary core north of certain latitude shows a particular deglacial behavior compared to the ones, which are located at higher latitudes?

Reply #5: Please see reply #4 above for an explanation as to why authigenic Pb and Nd isotope records show different trends. We expanded the section discussing South Atlantic cores slightly for setting the changes seen at our site better in context with existing records. We are the first to present a reliable record from a deep marine core site so far south in the Atlantic sector of the SO. From earlier records (Lippold et al., 2016) the lack of an early deglacial AABW export was not clearly apparent, although traceable, when directly comparing the ODP Site 1089 ϵ_{Nd} record with that of Site PS1768-8 (see Figs. 2g, h). We further clearly flag the section in our core preceding the onset of Weddell Sea-derived AABW arrival as controlled by porewater processes, a phenomenon that was increasingly discussed in recent years (Abbott et al., 2015; Du et al., 2016; Haley et al., 2017). That said, we reiterate that under active deep water circulation the extracted bottom water ϵ_{Nd} signal is deemed reliable (Blaser et al., 2019). Besides the good agreement between coretop ϵ_{Nd} with nearby SO deep water sampling locations, support for this claim also comes from a shelf seawater study in the Arctic Laptev Sea (Laukert et al., 2017). Usually the nature of the marine sediments will determine the reliability of extracted authigenic Nd isotope compositions. Antarctic sediments that were only glacially eroded commonly have chemically reactive surfaces, which may potentially release trace metals after deposition, thereby altering the porewater signature. This is the reason why a coretop comparison to ambient bottom water compositions is mandatory in settings like here.

The authors discussed the integrity of their down core Nd isotope data. I am very much in favor of the approach of using pure opal to generate Nd isotope record, it might be the only way if high-resolution records are to be generated for high latitude Southern Ocean. Being said that, the logic presented in the manuscript to establish that their records are indeed representative of bottom water is a bit circular. In the supplementary section Fe-Mn oxyhydroxide derived eNd isotope from core PS 1768 are presented along with a few opal-derived data. While the opal derived eNd numbers matched with downcore data, it is necessary to have a pure opal derived value for the core top. Should the core top Fe-Mn oxyhydroxide, core top pure opal, and ambient bottom water eNd values match, then it would establish beyond doubt that the generated data is not an artifact of their method. Another approach could have been to compare the carbonate content of the core and Fe-Mn oxyhydroxide derived eNd record. If there is high correlation between how the carbonate content changed in the core and how the downcore eNd changed, then it would be difficult to argue that Fe-Mn oxyhydroxide derived data are pure bottom water.

Reply #6: Agreed. We analysed the available core top (4cm from the surface) opal Nd isotope composition and the result matches the Nd isotope ratio in Fe-Mn oxyhydroxide and ambient bottom water. The data has been added into the revised manuscript and is included in the Supplementary Figure 1.

In line 137, the authors claimed 'The authigenic signal is also different from and does not co-vary with the detrital eNd signatures...'. I agree that the authigenic values are different but it is unclear if they co-vary. The detrital record is low resolution and the overall pattern of change in the detrital record has a resemblance to the Fe-Mn oxyhydroxide record for the last ~20 kyr.

Reply #7: Figure R1 below illustrates the lack of covariation between the authigenic and detrital Nd isotopic signature. The sentence in the revised manuscript has been changed to 'The authigenic signal is also different from the detrital Nd isotope signatures' (now Line 195-197).

Figure R1. Lack of correlation between the authigenic and detrital Nd isotope signature in ϵ_{Nd} - ϵ_{Nd} space. If the authigenic and detrital fraction were co-varying, we would expect a significantly higher regression coefficient, which is not the case. Also shown is the 1:1 line.

The downcore ϵ_{Nd} record in core PS1768-8 has values around -4, which the authors have interpreted to be due to sedimentary boundary release processes. They further claimed that it is only prevalent during times of sluggish circulation i.e during glacial and early deglacial times. However, if this is true, then is it not possible that any changes in the downcore Nd record that could have been due to AABW export during the glacial and early deglacial times might have been overprinted by sedimentary processes?

Reply #8: Please see also reply #5 above regarding the discussion about porewater processes and their potential impact on the Nd isotope signal. The change in ϵ_{Nd} at ~ 13 ka is very pronounced, and switching from a highly radiogenic (porewater-controlled) composition to the characteristic AABW composition seen in the Atlantic sector of the SO. In theory, if AABW had been advected to the core site under very sluggish conditions during earlier parts of the deglaciation, we indeed would have missed its arrival. We cannot entirely exclude this possibility although we consider it very unlikely since onset of a stronger bottom water circulation regime should equally result in authigenic ϵ_{Nd} that are more representative of the overlying bottom water signal (which is not observed). However, as outlined in the manuscript

(lines 197-199), the very clear-cut onset of AABW presence at Site PS1768-8 calls for a major deep water circulation change, since such a sharp transition would otherwise not be preserved. Besides, if there had been active AABW export event before 13 ka, the ϵ_{Nd} at Site PS1768-8 should show similar compositions with that in PS1599-3 such as seen after the late deglacial (Fig. 2g).

It is also not clear to me how the authors arrived at the conclusion of no AABW export during the glacial. The glacial part of the record in PS 1768-8 is really short (and does not even cover LGM) and moreover, it is claimed to be overprinted by sedimentary processes. Is this conclusion based on Pb isotopes? If so then there needs to be a clear explanation. A water mass mixing exercise to calculate the relative proportion of AABW and other relevant water masses might be useful.

Reply #9: The reviewer is partially correct. Our oldest Nd isotope data point during T1 dates back to 19 ka, which strictly speaking corresponds to the end of the LGM as outlined in Clark et al. (2009). On the other hand, the Nd isotope record during TII extends into the Penultimate Glacial Maximum (PGM) and no change is apparent. Furthermore, the Pb and Nd isotope records during the PGM are very similar to those seen during the late LGM (Fig. 5). Given that the Pb isotope composition during the LGM is remarkably invariant compared with the earliest deglacial section preceding 18 ka, we consider it highly unlikely that the circulation regime in the Atlantic sector of the SO was significantly different from the earliest deglacial configuration.

I am supportive of this manuscript with a major revision.

Reply #10: We thank reviewer #1 for her/his overall positive verdict.

Reviewer #2 (Remarks to the Author):

This paper uses lead isotopes and ϵ_{Nd} (epsilon Nd) to document the changing contribution of Weddell Sea bottom water to the South Atlantic sector of the Southern Ocean over the last two glacial terminations. In this regard it is similar to Basak et al. (Nature, 359, 900-904, 2018) who used changes in ϵ_{Nd} to look at the contribution of Ross Sea bottom water to the Pacific sector of the Southern Ocean. The problem is that the ϵ_{Nd} story in the present paper is not nearly as clear-cut as it is in the Pacific. The Pb isotope story, on the other hand, seems to be very informative. Hence, my recommendation is that the authors jettison the ϵ_{Nd} data in the paper and focus instead on the Pb isotope data.

Reply #11: We thank reviewer #2 for her/his constructive comments. The Pb and Nd story in our manuscript indeed share some resemblances with the mentioned recent paper (Basak et al., 2018) but the main paleoceanographic implications of our record are clearly different from the earlier publication. The reviewer is right in that the deglacial part of our Nd record is more difficult to interpret than corresponding records in the Pacific sector of the SO (Basak et al., 2018). However, given the behavior of Nd in extreme deep marine settings (see reply #5 above) even an ϵ_{Nd} that

is offset from expected bottom water compositions in early deglacial parts of the record can provide valuable insights into paleoceanographic configurations. Since we clearly address and discuss the exotic signatures seen during most of the deglaciation and can provide a previously reported mechanism capable of generating such compositions, we are convinced that the Nd isotope record holds vital information that should be included in the manuscript (see also reply #8 above).

General readers, like myself, could use some more background on the Pb isotopes in the South Atlantic. The authors obviously went to some trouble to compile Pb isotope compositions from dozens of core tops across a large area (Fig. 2) and they found a wide range of compositions that reflect on the downcore compositions at ODP site 1094. The comparison shows that the deep water at site 1094 seems to have been dominated by relatively unradiogenic water with origins in the Pacific or South Atlantic during the last two glacial periods. But then more radiogenic water began to be swept out of the Weddell Sea starting about 18,000 and 135,000 years ago. The implication is that the more radiogenic compositions were confined to the Weddell Sea during the prior glacial periods because there was no production of Weddell Sea bottom water to carry these compositions out to the larger area beyond. This is the most instructive finding in the paper. But how robust is it? Why are the Pb isotope compositions so much more radiogenic in the eastern and western Weddell Sea in relation to Drake Passage and the SE Atlantic?

Reply #12: See also replies #1 and #4. In general, Pb isotope compositions are more variable compared with Nd isotopes because of the shorter residence time of Pb in seawater compared with ϵ_{Nd} alongside more isotopic variability in continental crust. Furthermore, Pb has three radiogenic isotopes derived from two mother nuclides (^{238}U , ^{235}U and ^{232}Th), which together provide a much more diagnostic source signature compared with Nd. Seawater Pb is mainly supplied by continental runoff so its Pb isotope signature in the ocean depends on the composition and age of the weathered rock sources (Frank et al., 2002). Consequently, the much more radiogenic Pb isotope compositions in the eastern and western Weddell Sea in contrast to Drake Passage and the SE Atlantic are controlled by corresponding crustal source signatures underneath the Antarctic ice shield. Given the clear spatial difference in authigenic Pb isotope signatures between the eastern and west Weddell Sea we can even roughly distinguish water mass sources from the Western or Eastern Weddell Sea (Fig. 3).

Reviewer #3 (Remarks to the Author):

Review of Huang et al.

Huang et al. investigate changes in Weddell Sea derived Antarctic Bottom water across the last and penultimate deglaciation, and the Eemian. For this they present Pb isotope records from Southern Ocean core ODP1094, as well as Nd isotope records from nearby core PS1768-8 and Weddell Sea core PS1599-3. ODP1094 and

PS1768-8 cores cover the last deglaciation (~20-7ka) as well as the penultimate deglaciation and the Eemian (~143-115 ka), while core PS1599-3 only covers the last deglaciation.

Given the importance of AABW for the global carbon cycle, gaining a better understanding on sources and temporal changes in AABW across both deglaciations and the Eemian is crucial. I think this study does make a step in this direction by presenting original and relatively high-resolution records from the Southern Ocean. So, in principle I would support the study for publication in Nature Communications. However, below are some comments that would first need to be addressed.

Reply #13: We also thank reviewer #3 for detailed and insightful comments on the earlier version of our manuscript.

1) Interpretation of Pb isotopes

The ϵ_{Nd} record of PS1599-3 in the Weddell Sea show little to no change across the deglaciation and the ϵ_{Nd} record from PS1788-8 reaches the value of PS1599 rather rapidly. The authors interpret this as reflecting the dominance of Weddell Sea waters, which formation could have been quite influenced by the ice-sheet retreat over the Weddell Sea.

Reply #14: See also replies #4 and 5 for the underlying controls of our Nd isotope record at Site PS1768-8. The reviewer is likely correct that the relatively abrupt onset of Weddell Sea-derived AABW export likely initiated as a function of ice-sheet retreat over the Weddell Sea. However, to date reconstructions on sea ice extent during the deglaciation in the Weddell Sea are still rare (e.g. Xiao et al., 2016) and associated with large uncertainties, for which reason we do not want to include a major discussion on this issue. In fact, a sub-mesoscale state-of-the-art ocean circulation model would be required to reconstruct feasible deglacial sea ice retreat pattern, since AABW formation today occurs on the Weddell Sea shelf (e.g. Purkey et al., 2018), a process which most oceanic coarser-resolution models to date cannot resolve. We do make a reference to MWP1A in the revised version of the manuscript, yet in context of Pb isotopic changes (lines 160-164).

The Pb isotopes records from ODP1094 are quite interesting as they could reflect less local changes. I should flag that this is the first time I come across Pb isotopes records, so I have no knowledge on this proxy. Given the authors' explanation of the proxy, the data shown in Figure 2 and the records, I can be convinced that the increase in $^{208}/^{204}\text{Pb}$ and $^{206}/^{204}\text{Pb}$ could be due to an increased proportion of Western Weddell Sea waters.

However, the authors write L.98-99, that the Pb changes are due to a southward displacement of the front as seen in Xiao et al. Here, we are not talking about fronts, but about the location of formation, pathway (and strength?) of "southern-sourced deep waters".

Reply #15: As outlined in reply #4 above, the extracted authigenic Pb isotope signatures monitor the average composition of the entire water column, not the bottom water signature. Our Pb isotope record hence indeed monitor changes in the

upper water column (further discussion in reply #4). Only the small change seen at the onset of Weddell Sea-derived AABW export after 13 ka (Fig. 2b, c) was most likely advected to the core site within newly formed AABW.

In addition, it would be good to know the Pb isotopes signature of NADW (I could not find it). This is particularly important since ODP1094 is not that deep. Maybe some additional information on Pb isotopes could be added to the supplementary information.

Reply #16: The natural modern seawater Pb isotope signature can no longer be quantified directly because of the anthropogenic Pb contamination (Flegal et al., 1993; Lee et al., 2015). To circumvent this problem of a missing modern reference frame, the pre-industrial seawater Pb signatures can be extracted from Fe-Mn nodules or authigenic Fe-Mn oxyhydroxides in sediments such as done here (Fig. 3). The Pb isotope signature of NADW has been reported in Klemm et al. (2007). Including the Pb isotope composition of NADW in a SO paper is pointless, however, since North Atlantic Pb would be scavenged from the water column long before NADW reaches the SO (Frank, 2002). Dissolved Pb in NADW water depths at SO sites would hence be replenished from more proximal sources (South America, Africa, Antarctic Peninsula etc.). This is another reason why we show the Pb isotope variability in the Atlantic sector of the SO via own new coretop authigenic Pb isotope compositions alongside Fe-Mn nodule data of Abouchami and Goldstein (1995) in Fig. 3.

2) Scope of the study

The study presents records covering both deglaciation and the Eemian. One could argue that this could be split into 2 or 3 papers. I understand why the authors would want to show everything in one paper, but then the scope of the study becomes quite large and the details given in the different topics become quite small. Additional information and details on the LGM and the last deglaciation could be given if the study was to focus on that. It would make the study clearer. The penultimate deglaciation is only briefly mentioned with no background information at all. The Eemian is rushed through...

Reply #17: We understand the notion of reviewer #3 to focus on each termination separately, yet the approach of comparing glacial terminations in a single paper is not uncommon. From our point of view, showing the temporal evolution of our proxies side by side for TI and TII is the most practical way, both to highlight similarities but also key differences. Following the advice of the editor (see reply #1) we decided to leave the TII record in the revised version of our manuscript. In order to provide a better introduction to the overall climatic evolution, we added more background information during TII (lines 225-232).

3) Introduction

The introduction would need to be more precise and associated references more appropriate.

- L.35-36: “pCO₂ increased in 2 main intervals”: please state which intervals you are referring to. Also, more recent higher resolution atmospheric CO₂ records have been published: e.g. Marcott et al. 2014 Nature.

Reply #18: “pCO₂ increased in 2 main intervals” is revised as “atmospheric pCO₂ increased during two main intervals between 18-15 ka and 13-11 ka (Monnin et al., 2004) (Fig. 2a), which occurred during Southern Hemispheric warming phases”. We also included the atmospheric pCO₂ record of (Marcott et al., 2014) as requested.

- L. 37-38: I am not really sure “a consensus could be reached” that it was due to a “southward moving overturning cell”. Changes in the southern hemispheric westerlies have been proposed (2,8), and they would have had an impact on the SO overturning cell and atmospheric CO₂ (see also Menviel et al., 2018), pointing to a strengthening of the SO overturning, but whether it was shifted southward is up to debate, and in fact this is where your study is quite important.

Reply #19: Sentence revised, now also acknowledging the recent finding of Marcott et al. (2014) that three minor yet resolvable steps in atmospheric pCO₂ rise likely have their origin in Northern Hemisphere climatic events (now lines 35-38).

We further thank the reviewer for mentioning the importance of our data set for the glacial-interglacial reconstructions of SO fronts. As correctly stated by the reviewer, although southward-shifted westerlies in the southern hemispheric have been proposed in past studies (Toggweiler, 1999), the degree of shifting of SO fronts is still under debate, and only few studies provided information in this direction (e.g. Xiao et al., 2016). We also thank the reviewer for pointing out the Menviel et al. (2018) study to us, which is now referenced in line 34.

- L. 39: I don’t think ref. 6 is appropriate here. Ref. 2 already points to a stronger upwelling during the deglaciation.

Reply #20: Fair point. The Hasenfratz et al. (2019) study is a really nice piece of work but not entirely appropriate in this context since the authors mostly discuss SO surface/sub-surface processes. We removed the reference in the revised manuscript.

- L. 39-42: This paragraph is really confusing, particularly in the context on what was discussed above. To which deglacial pulse are you referring to? “The bipolar seesaw” already operated during Heinrich 1, and is a term that means nothing and everything. Why would you discuss AMOC reductions only for “the second pulse”? The strongest AMOC reduction during the deglaciation most likely occurred during Heinrich 1 (e.g. Ng. et al., 2018).

Reply #21: Agreed. Sentence deleted.

4) Weddell Sea waters and SO overturning

This comment refers to L. 129-132 in particular, but it also refers to a comment made above, and would push for a more precise use of terms throughout the manuscript.

The precursor of AABW today are dense shelf waters (DSW) formed on the Antarctic continental shelves by brine rejection. The main regions of formation are the

Weddell and Ross Seas. While the authors are completely correct that i) the location of formation of DSW could have changed in the past, ii) there is no study particularly looking at how the sources changed in the past, several studies have already pointed out that due to the Antarctic ice-sheet advance in the Ross and Weddell Seas, AABW formation might have been quite different in the past (e.g. Paillard & Parrenin, 2004, EPSL, Menviel et al., 2017). In their discussion, the authors should take into account the Antarctic ice-sheet advance (and subsequent retreat) over the Weddell Sea (Golledge et al., 2014 present a potential evolution).

Reply #22: Regarding terminology of regional water masses, we are aware that the individual water masses within the Weddell Sea have a whole range of names, beginning with the difference between Weddell Sea Bottom and Weddell Sea Deep Water but extending far beyond such a distinction (Huhn et al., 2008). Since we can only report within this manuscript what arrived at ODP Site 1094/PS1788-8, we think using the generic term “Weddell Sea-derived AABW” is most appropriate (strictly speaking this would be Weddell Sea Deep Water admixed into Lower Circumpolar Deep Water within the Scotia Sea (Naveira Garabato et al., 2002). We included a separate paragraph in lines 52-59 within the revised manuscript to briefly introduce the water column structure within the Weddell Sea.

We thank the reviewer for bringing the Golledge et al. (2014) study to our attention, which indeed provides a striking additional angle on the evolution of our deglacial Pb isotope record. As outlined in their study (as well as in Mackintosh et al., 2014), the effect of excessive Antarctic melting during meltwater pulse 1a (MWP1a) is highlighted. This timing corresponds to the onset of the Antarctic Cold Reversal, which was suggested to result in reduced SO overturning circulation (Anderson et al., 2009; Skinner et al., 2010). Following the trend towards more radiogenic compositions between 18 and 15 ka at ODP Site 1094, our Pb isotope records change towards slightly less radiogenic compositions during this time interval (Fig. 2b,c), which may be a direct reflection of increased Antarctic meltwater input during this time interval and its effect on SO overturning dynamics. This comment also made us revisit another paper published a few years ago that presented evidence for highest iceberg rafted debris deposition in the Scotia Sea during MWP1A (Weber et al., 2014). Both studies agree with our findings and are now mentioned in lines 160-164.

In addition, it is quite possible that the AABW transport changed significantly. While I agree that AABW transport at the LGM could have been reduced, I am not sure this view would fit as “commonly” (L. 131), and I am not sure that the references included in that sentence are all appropriate. Inclusion of Ref. 1 could be discussed. Ref. 2 nicely infers a stronger Southern Ocean upwelling during Heinrich 1, however there is no indication of AABW formation at the LGM in their study.

Reply #23: Agreed. “Commonly” changed to “In prior studies” since not many studies exist that provided data from near-Antarctic settings regarding the status of AABW during past climates. We further deleted references 1 (Sigman and Boyle 2000) and 2 (Anderson et al. 2009) from this statement and included a reference to Smith et al.

(2010), which is more appropriate.

-L. 195-196: “too dense to be exported to our northern SO core site”. I understand that under special circumstances (and in modern day), if the “water is too dense” it could be blocked by topographic features, however in your case I don’t think there is any evidence for such a mechanism at play. The authors might want to consider here the ice-sheet advance over the Weddell Sea (Golledge et al., 2014, Bentley et al., 2014) at glacial maxima. (L. 202-205) Given the location of your cores, an eventual blockage on the northern route should not matter.

Reply #24: We disagree on this point. Although the Antarctic ice shelf edge likely did not extend beyond the Weddell Sea shelf during the LGM (Bentley et al., 2014; Hillenbrand et al., 2014), it is not clear whether conditions were favourable at all for new AABW formation, or whether deep water formation rates were for example reduced. Today, newly formed Weddell Sea AABW is formed at various locations in the Weddell Sea as Weddell Sea Bottom Water. This water mass is transformed into the warmer and less dense Weddell Sea Deep Water above via active help of the overlying Weddell Gyre that brings Warm Deep Water into the southern reaches of the Weddell Sea (Huhn et al., 2008; Kerr et al., 2018; Reeve et al., 2019). Weddell Sea Deep Water is exported to the north into the Scotia Sea, where it is effectively mixed with Lower Circumpolar Deep Water (Naveira Garabato et al., 2002) before reaching our core sites in the Atlantic sector of the SO. The northern edge of the Weddell Sea is bound by a bathymetric high, the South Scotia Ridge and the Bouvet Triple Junction. Hence if deep water formation was ongoing yet reduced during the LGM, newly formed WSBW may have not been as effectively mixed upward into Weddell Sea Deep Water, hence not being able to pass this bathymetric high in transit to the Scotia Sea under reduced glacial Weddell Gyre circulation. Since unblocked export of Weddell Sea AABW would still be possible to the northeast towards the Indian sector of the SO into the Weddell-Enderby Abyssal Plain (no bathymetric restriction) and no data exist from this part of the Southern Ocean, we left the possibility open that glacial AABW formed in the Weddell Sea during the LGM could in theory still be exported (see Fig. 1 in Hellmer et al., 2016).

Figure 2: It would be easier for the reader if the y axis was $^{208}\text{Pb}/^{204}\text{Pb}$, as shown for core ODP1094 in figures 1 and 4.

Reply #25: We tested the use of $^{208}\text{Pb}/^{204}\text{Pb}$ as y axis but Pb sources from different areas in $^{206}\text{Pb}/^{204}\text{Pb}$ - $^{208}\text{Pb}/^{204}\text{Pb}$ isotope space are not as distinct as in $^{206}\text{Pb}/^{204}\text{Pb}$ - $^{208}\text{Pb}/^{206}\text{Pb}$ in Fig. 2 (now Fig. 3). This becomes evident in $^{206}\text{Pb}/^{204}\text{Pb}$ - $^{208}\text{Pb}/^{204}\text{Pb}$ isotope space in Extended Data Fig. S6.

Figure 3: I understand that the authors want to include a diagram, but in its present form it is not possible to show this figure as several aspects are not substantiated. Given the text and records, the authors are really looking at the Weddell Sea, therefore the figures should really reflect that this is the Weddell Sea deep waters that are represented here and not the “Southern Ocean”.

Then several aspects of the figure could be discussed/argued: i) HS1, YD are not defined nor really discussed in the text, ii) where does the sea-ice edge comes from? I can understand that the LGM sea-ice edge is further than the c) one, but what about b)? iii) where does the changes in the CDW arrow come from? iv) where do the water mass mixing changes come from? v) whereas the study mostly reflect changes in the source of AABW it is referred in the same way in the 3 diagrams, the only difference being "AABW going over the hump in c". Given the position of the cores, I'm not sure what the "hump" refers to.

Finally, one could argue about the relative depth positions of the cores with respect to their real positions.

Reply #26: This diagram not only sketches processes in the Weddell Sea, our records have paleoceanographic implications for the Weddell Sea and the Atlantic sector of the SO. The purpose of this diagram is to illustrate the different controls over seawater-derived Pb and Nd isotopes as well as to show the underlying oceanic processes that change their value.

Following to the reviewer's comment, 1) HS1, YD are introduced in the caption of Fig. 2. While the term (post-)YD is also used for Fig. 3c, we replaced "HS1" with "preceding the ACR (~15 ka)", which is a more accurate description of the time window shown here 2) the sea-ice edge information is entirely schematic and reflects the deglacial retreat of sea ice as a feedback of increasing temperature and similar figures can be found in previous studies; 3) the southward moving upper circulation cell leading to upwelling further south serves to illustrate the processes at work during the early deglacial Pb isotopic shifts in our records. As a consequence of the southward shifting SO overturning cell, increasing contributions of a Weddell Sea-sourced radiogenic (high) Pb isotope signal into CDW leads to an increase $^{206}\text{Pb}/^{204}\text{Pb}$ in the surface water above the core site outside the Weddell Sea at ODP Site 1094. 4) The hump is related to the Bouvet Triple Conjunction Ridge (the extension of the South Scotia Ridge) as also shown in new Fig.1 of our revised manuscript (see also reply #24). We now also clearly link the individual sub-figures of Fig. 4 to into the discussion.

Minor points and typos:

- L. 76: Core PS1599 is at 27 degrees West (and not East)!

Reply #27: Corrected.

- CDW is not defined (and nor are UCDW and LCDW later in the text)

Reply #28: CDW as well as UCDW and LCDW are now defined in lines 91, 131 and 206 in the revised manuscript.

- L. 81-82: I am not sure what the authors define here as "upper and lower SO overturning cells", and not sure that "the records resolve changes in the upper and lower cells". I would agree with the "lower cell" as changes in AABW composition, but I can't see where the "upper one" comes in.

Reply #29: As explained above in reply #4, Pb mostly monitored changes in the upper SO overturning circulation cell until late during TI and TII, while the Nd isotope record traced abyssal circulation trends in the lower circulation cell.

- L. 153: “where”

Reply #30: This process has been reported for the North Pacific and the North Atlantic, information and references have been added in Lines 210.

- L. 163: I am not sure the changes the authors are seeing are due to changes in insolation and the term “Milankovitch forcing” might not be appropriate here.

Reply #31: We appreciate this comment, but ultimately climatic and oceanographic changes on glacial-interglacial timescales should be driven by changing incoming solar insolation that is controlled by orbital parameters, so we would think the term “Milankovitch forcing” is appropriate.

References used in replies

Abbott, A.N., Haley, B.A., McManus, J., 2015. Bottoms up: Sedimentary control of the deep North Pacific Ocean’s ϵ Nd signature. *Geology* 43, 1035-1035.

Abouchami, W., Goldstein, S.L., 1995. A lead isotopic study of circum-Antarctic manganese nodules. *Geochimica et Cosmochimica Acta* 59, 1809-1820.

Anderson, R.F., Ali, S., Bradtmiller, L.I., Nielsen, S.H.H., Fleisher, M.Q., Anderson, B.E., Burckle, L.H., 2009. Wind-driven upwelling in the Southern Ocean and the deglacial rise in atmospheric CO₂. *Science* 323 1443-1448.

Basak, C., Fröllje, H., Lamy, F., Gersonde, R., Benz, V., Anderson, R.F., Molina-Kescher, M., Pahnke, K., 2018. Breakup of last glacial deep stratification in the South Pacific. *Science* 359, 900-904.

Bentley, M.J., Cofaigh, C., Anderson, J.B., Conway, H., Davies, B., Graham, A.G.C., Hillenbrand, C.-D., Hodgson, D.A., Jamieson, S.S.R., Larter, R.D., Mackintosh, A., Smith, J.A., Verleyen, E., Ackert, R.P., Bart, P.J., Berg, S., Brunstein, D., Canals, M., Colhoun, E.A., Crosta, X., Dickens, W.A., Domack, E., Dowdeswell, J.A., Dunbar, R., Ehrmann, W., Evans, J., Favier, V., Fink, D., Fogwill, C.J., Glasser, N.F., Gohl, K., Golledge, N.R., Goodwin, I., Gore, D.B., Greenwood, S.L., Hall, B.L., Hall, K., Hedding, D.W., Hein, A.S., Hocking, E.P., Jakobsson, M., Johnson, J.S., Jomelli, V., Jones, R.S., Klages, J.P., Kristoffersen, Y., Kuhn, G., Leventer, A., Licht, K., Lilly, K., Lindow, J., Livingstone, S.J., Masson, G., McGlone, M.S., McKay, R.M., Melles, M., Miura, H., Mulvaney, R., Nel, W., Nitsche, F.O., O'Brien, P.E., Post, A.L., Roberts, S.J., Saunders, K.M., Selkirk, P.M., Simms, A.R., Spiegel, C., Stollendorf, T.D., Sugden, D.E., van der Putten, N., van Ommen, T., Verfaillie, D., Vyverman, W., Wagner, B., White, D.A., Witus, A.E., Zwart, D., 2014. A community-based geological reconstruction of Antarctic Ice Sheet deglaciation since the Last Glacial Maximum. *Quaternary Science Reviews* 100, 1-9.

Blaser, P., Pöppelmeier, F., Schulz, H., Gutjahr, M., Frank, M., Lippold, J., Heinrich, H., Link, J.M., Hoffmann, J., Szidat, S., Frank, N., 2019. The resilience and sensitivity of Northeast Atlantic deep water ϵNd to overprinting by detrital fluxes over the past 30,000 years. *Geochimica et Cosmochimica Acta* 245, 79-97.

Clark, P.U., Dyke, A.S., Shakun, J.D., Carlson, A.E., Clark, J., Wohlfarth, B., Mitrovica, J.X., Hostetler, S.W., McCabe, A.M., 2009. The Last Glacial Maximum. *Science* 325, 710-714.

Du, J., Haley, B.A., Mix, A.C., 2016. Neodymium isotopes in authigenic phases, bottom waters and detrital sediments in the Gulf of Alaska and their implications for paleo-circulation reconstruction. *Geochimica et Cosmochimica Acta* 193, 14-35.

Ferrari, R., Jansen, M.F., Adkins, J.F., Burke, A., Stewart, A.L., Thompson, A.F., 2014. Antarctic sea ice control on ocean circulation in present and glacial climates. *Proceedings of the National Academy of Sciences* 111, 8753-8758.

Flegal, A.R., Maring, H., Niemeier, S., 1993. Anthropogenic lead in Antarctic sea water. *Nature* 365, 242.

Frank, M., 2002. RADIOGENIC ISOTOPES: TRACERS OF PAST OCEAN CIRCULATION AND EROSIONAL INPUT. *Reviews of Geophysics* 40, 1-1-1-38.

Frank, M., Whiteley, N., Kasten, S., Hein, J.R., O'Nions, K., 2002. North Atlantic Deep Water export to the Southern Ocean over the past 14 Myr: Evidence from Nd and Pb isotopes in ferromanganese crusts. *Paleoceanography* 17, 12-11-12-19.

Golledge, N.R., Menviel, L., Carter, L., Fogwill, C.J., England, M.H., Cortese, G., Levy, R.H., 2014. Antarctic contribution to meltwater pulse 1A from reduced Southern Ocean overturning. *Nature Communications* 5, 5107.

Haley, B.A., Du, J., Abbott, A.N., McManus, J., 2017. The Impact of Benthic Processes on Rare Earth Element and Neodymium Isotope Distributions in the Oceans. *Frontiers in Marine Science* 4.

Hellmer, H.H., Rhein, M., Heinemann, G., Abalichin, J., Abouchami, W., Baars, O., Cubasch, U., Dethloff, K., Ebner, L., Fahrbach, E., Frank, M., Gollan, G., Greatbatch, R.J., Grieger, J., Gryanik, V.M., Gryschka, M., Hauck, J., Hoppema, M., Huhn, O., Kanzow, T., Koch, B.P., König-Langlo, G., Langematz, U., Leckebusch, G.C., Lüpkes, C., Paul, S., Rinke, A., Rost, B., van der Loeff, M.R., Schröder, M., Seckmeyer, G., Stichel, T., Strass, V., Timmermann, R., Trimborn, S., Ulbrich, U., Venchiarutti, C., Wacker, U., Willmes, S., Wolf-Gladrow, D., 2016. Meteorology and oceanography of the Atlantic sector of the Southern Ocean—a review of German achievements from the last decade. *Ocean Dynamics* 66, 1379-1413.

Hillenbrand, C.-D., Bentley, M.J., Stollendorf, T.D., Hein, A.S., Kuhn, G., Graham, A.G.C., Fogwill, C.J., Kristoffersen, Y., Smith, J.A., Anderson, J.B., Larter, R.D., Melles, M., Hodgson, D.A., Mulvaney, R., Sugden, D.E., 2014. Reconstruction of changes in the Weddell Sea sector of the Antarctic Ice Sheet since the Last Glacial Maximum. *Quaternary Science Reviews* 100,

111-136.

Huhn, O., Hellmer, H.H., Rhein, M., Rodehacke, C., Roether, W., Schodlok, M.P., Schroeder, M., 2008. Evidence of deep- and bottom-water formation in the western Weddell Sea. *Deep-Sea Research Part II-Topical Studies in Oceanography* 55, 1098-1116.

Kerr, R., Dotto, T.S., Mata, M.M., Hellmer, H.H., 2018. Three decades of deep water mass investigation in the Weddell Sea (1984-2014): Temporal variability and changes. *Deep-Sea Research Part II-Topical Studies in Oceanography* 149, 70-83.

Klemm, V., Reynolds, B., Frank, M., Pettke, T., Halliday, A.N., 2007. Cenozoic changes in atmospheric lead recorded in central Pacific ferromanganese crusts. *Earth and Planetary Science Letters* 253, 57-66.

Laukert, G., Frank, M., Bauch, D., Hathorne, E.C., Gutjahr, M., Janout, M., Hölemann, J., 2017. Transport and transformation of riverine neodymium isotope and rare earth element signatures in high latitude estuaries: A case study from the Laptev Sea. *Earth and Planetary Science Letters* 477, 205-217.

Lee, J.-M., Boyle, E.A., Gamo, T., Obata, H., Norisuye, K., Echegoyen, Y., 2015. Impact of anthropogenic Pb and ocean circulation on the recent distribution of Pb isotopes in the Indian Ocean. *Geochimica et Cosmochimica Acta* 170, 126-144.

Lippold, J., Gutjahr, M., Blaser, P., Christner, E., de Carvalho Ferreira, M.L., Mulitza, S., Christl, M., Wombacher, F., Böhm, E., Antz, B., Cartapanis, O., Vogel, H., Jaccard, S.L., 2016. Deep water provenance and dynamics of the (de)glacial Atlantic meridional overturning circulation. *Earth and Planetary Science Letters* 445, 68-78.

Mackintosh, A.N., Verleyen, E., O'Brien, P.E., White, D.A., Jones, R.S., McKay, R., Dunbar, R., Gore, D.B., Fink, D., Post, A.L., Miura, H., Leventer, A., Goodwin, I., Hodgson, D.A., Lilly, K., Crosta, X., Golledge, N.R., Wagner, B., Berg, S., van Ommen, T., Zwartz, D., Roberts, S.J., Vyverman, W., Masse, G., 2014. Retreat history of the East Antarctic Ice Sheet since the Last Glacial Maximum. *Quaternary Science Reviews* 100, 10-30.

Marcott, S.A., Bauska, T.K., Buizert, C., Steig, E.J., Rosen, J.L., Cuffey, K.M., Fudge, T.J., Severinghaus, J.P., Ahn, J., Kalk, M.L., McConnell, J.R., Sowers, T., Taylor, K.C., White, J.W.C., Brook, E.J., 2014. Centennial-scale changes in the global carbon cycle during the last deglaciation. *Nature* 514, 616.

Monnin, E., Steig, E.J., Siegenthaler, U., Kawamura, K., Schwander, J., Stauffer, B., Stocker, T.F., Morse, D.L., Barnola, J.-M., Bellier, B., Raynaud, D., Fischer, H., 2004. Evidence for substantial accumulation rate variability in Antarctica during the Holocene, through synchronization of CO₂ in the Taylor Dome, Dome C and DML ice cores. *Earth and Planetary Science Letters* 224, 45-54.

Naveira Garabato, A.C., McDonagh, E.L., Stevens, D.P., Heywood, K.J., Sanders, R.J., 2002. On the export of Antarctic Bottom Water from the Weddell Sea. *Deep Sea Research Part II:*

Topical Studies in Oceanography 49, 4715-4742.

Reeve, K.A., Boebel, O., Strass, V., Kanzow, T., Gerdes, R., 2019. Horizontal circulation and volume transports in the Weddell Gyre derived : from Argo float data. *Progress in Oceanography* 175, 263-283.

Shakun, J.D., Clark, P.U., He, F., Marcott, S.A., Mix, A.C., Liu, Z., Otto-Bliesner, B., Schmittner, A., Bard, E., 2012. Global warming preceded by increasing carbon dioxide concentrations during the last deglaciation. *Nature* 484, 49.

Skinner, L.C., Fallon, S., Waelbroeck, C., Michel, E., Barker, S., 2010. Ventilation of the deep Southern Ocean and deglacial CO₂ rise. *Science* 328 1147-1151.

Talley, L.D., 2013. Closure of the Global Overturning Circulation Through the Indian, Pacific, and Southern Oceans: Schematics and Transports *Oceanography* 26, 80-97.

Toggweiler, J.R., 1999. Variation of atmospheric CO₂ by ventilation of the ocean's deepest water. *Paleoceanography* 14, 571-588.

Weber, M.E., Clark, P.U., Kuhn, G., Timmermann, A., Sprenk, D., Gladstone, R., Zhang, X., Lohmann, G., Menviel, L., Chikamoto, M.O., Friedrich, T., Ohlwein, C., 2014. Millennial-scale variability in Antarctic ice-sheet discharge during the last deglaciation. *Nature* 510, 134-138.

Wu, J., Rember, R., Jin, M., Boyle, E.A., Flegal, A.R., 2010. Isotopic evidence for the source of lead in the North Pacific abyssal water. *Geochimica et Cosmochimica Acta* 74, 4629-4638.

Xiao, W.S., Esper, O., Gersonde, R., 2016. Last Glacial - Holocene climate variability in the Atlantic sector of the Southern Ocean. *Quaternary Science Reviews* 135, 115-137.

Reviewers' comments:

Reviewer #1 (Remarks to the Author):

Huang et al. in their revision of the manuscript 'No Weddell Sea Antarctic Bottom Water export during the Last and Penultimate Glacial Maximum' have addressed many of the key points raised by all three reviewers. The overall quality of the manuscript has improved significantly. However, I still have some nagging concerns which I cannot let go.

Comments:

Now that the authors have discussed in detail the proxies and how they intent to interpret Pb and Nd isotopes in their record, I have a few concerns that I would request them to address. According to what they have written, the following is my understanding

1. Pb isotopes in the water column are primarily controlled by Pb's particle reactive nature. However, according to their own admission (line 75) bottom water control is also present. So, to summarize, Pb isotopes extracted from Fe-Mn oxides coating on bulk sediments are a combination of integrated upper water column and bottom water signature. (In this context upper water column is a loosely used term and has no depth connotation, which might be important especially if one is discussing upper and lower cells of MOC).

2. Neodymium isotopes extracted from Fe-Mn oxide coating off bulk sediments, on the other hand is strictly controlled by deep ocean water masses and their mixing proportion, a classical water mass tracer. However, the picture is not that rosy any more as there are reports of benthic flux of Nd perturbing bottom water Nd isotope signatures whereby the bottom signals are altered towards porewater values. This is particularly true when deep ocean circulation is sluggish. Thus, to summarize Nd isotopes (in theory) can be a combination of bottom water mass and porewater compositions.

I agree with the authors that these contrasting characters can work to extract meaningful information especially if the pattern of change of Pb and Nd time series matches, then it would indicate that Pb signals is dominantly bottom water controlled. In the event the pattern of change between Pb and Nd isotope is not coherent, the Pb isotope changes can be interpreted as changes in the upper water column. In my opinion, this argument only works when the influence of porewater on the Nd isotopes can be shown to have minimal to no effect. It is quite clear (radiogenic value ~ -4 that is not common for this part of the ocean) that the downcore Nd record is affected by porewater influence for a prolonged part of their T1 record. The authors in their rebuttal argued that if there has been a prominent oceanic signal, it would have shown in the record and the record would not have been as invariant as they observe. They put forward the stark change in Nd isotopes ~ 13 ka as the evidence that when oceanic processes take over, porewater influence is easily overwhelmed. I understand the line of reasoning that the authors have put forward, however, it is still not as convincing and hard to defend. The authors could acknowledge this point in their manuscript. They should consider incorporating their "reply#8" in the main text so that the readers are aware of the caveat in the interpretation.

Minor comments:

Line 55: Need a citation.

Line 234: Not really 'several' thousand years.

Figure 1: Salinity is unitless.

Reviewer #2 (Remarks to the Author):

Although I no longer have the original, the revised manuscript is easier to read and tells a much more coherent story about changes in the production of Weddell Sea Bottom Water during the last two deglaciations. My only comment this time around would be to encourage the authors to take a look at Sikes et al. (Enhanced $\delta^{13}\text{C}$ and $\delta^{18}\text{O}$ Differences Between the South Atlantic and South Pacific During the Last Glaciation: The Deep Gateway Hypothesis, *Paleoceanography*, 32, <https://doi.org/10.1002/2017PA003118>, 2017). Fig. 3 in the Sikes et al. paper documents changes in $d^{18}\text{O}$ and $d^{13}\text{C}$ with depth in the South Atlantic and South Pacific during the most

recent deglaciation. They find a pattern in the South Atlantic that is much like the one in the present manuscript: an early replacement of water with glacial characteristics above 2 km followed by a later replacement of the deep water below 2 km. The early replacement in Sikes et al. (2017) takes place during HS1 like the changes in the Pb isotopes in Huang et al., while the late replacement takes place during the YD like Huang et al.'s changes in ϵNd . The early upper replacement in Sikes et al. (2017) could easily be described by the southward displacement of major oceanic fronts (lines 120 and 121 in Huang et al.) while the late deep replacement could be explained by a resumption of Weddell Sea Bottom Water formation.

Reviewer #3 (Remarks to the Author):

2nd review of Huang et al.

In this new version of the manuscript Huang et al. have better detailed the use of Pb isotopes. Since I was not familiar with Pb isotopes, this "new explanation" raises questions with respect to the interpretation of the records.

The Pb record from PS1599, in the Weddell Sea, does not show any time variations, therefore cannot be used to infer changes in Weddell Sea water. The 2 records that are useful are Pb isotopes from PS1768 and Nd isotopes from ODP1094. Both are situated within the CDW. Following the authors' explanations Pb isotopes represent changes occurring over the whole water column, at that location within the Antarctic Circumpolar Current and the CDW, whereas Nd would represent changes in bottom water. I note that both cores are in the deep ocean (3300m and 2800m depth), but not the abyssal ocean. While I agree that Weddell Sea waters could have a significant influence on both records, I doubt that they only record a Weddell Sea signal given the mixing happening within the ACC and CDW.

As such the Pb record during TI is interpreted as indicating changes in the upper SO cell, which the authors suggest agree with intensified upwelling of CDW. The problem is that the $\epsilon ps(Nd)$ record does not show any variations until late in the deglaciation (i.e. $\sim 13ka$). The authors thus suggest that there was no changes in Weddell-sea derived AABW until that time. This raises several issues:

- i) If there was intensified CDW upwelling during the early deglaciation, as also suggested in Figure 4, shouldn't this be visible in the $\epsilon ps(Nd)$ record?

- ii) How can the authors distinguished between Weddell-sea derived AABW and other AABW sources from the $\epsilon ps(Nd)$ record?

- iii) How can this $\epsilon ps(Nd)$ record, right in the middle of the CDW be so different from other $\epsilon ps(Nd)$ records from the region: i.e. Piotrowski et al., 2005 and 2012, Skinner et al., 2013 (Geology), and even the CDW in the Pacific side (Basak et al., 2018)? Even though they are a bit northward, both MD07-3076 (Skinner et al., 2013), and TN057-21 (Piotrowski et al., 2012) are quite deep (3800 m and over 4000m depth respectively), and therefore the early $\epsilon ps(Nd)$ shift is most likely related to AABW changes.

Despite it being a valuable effort, I thus remain very skeptical about the $\epsilon ps(Nd)$ record of PS1768, as well as the link between these records and Weddell-sea derived AABW.

I also note that from figure 4, it looks like the $\epsilon ps(Nd)$ record starts to decrease about 1ka, and maybe not "several thousand years" (L.234) after the Pb record, and right during HS11. I am less aware of $\epsilon ps(Nd)$ record from the region and covering TII, but maybe this makes more sense than the variations recorded during TI.

Other points:

- The introduction lacks precision with respect to changes in atmospheric CO₂. For example L. 27: "In either deglacial atm. CO₂ rise scenario" \diamond no deglacial atm. CO₂ rise scenario are presented before. To which scenarios are you referring to?

- While I understand that schematics are very valuable, I do not find convincing evidence between what is presented in the manuscript and what is shown in figure 5.

No Weddell Sea Antarctic Bottom Water export during the Last and Penultimate Glacial Maximum

by

Huang Huang, Marcus Gutjahr, Anton Eisenhauer and Gerhard Kuhn

Reviewer #1

Huang et al. in their revision of the manuscript 'No Weddell Sea Antarctic Bottom Water export during the Last and Penultimate Glacial Maximum' have addressed many of the key points raised by all three reviewers. The overall quality of the manuscript has improved significantly. However, I still have some nagging concerns which I cannot let go.

Reply #1: We thank all three reviewers again for their constructive criticism and will answer remaining comments below.

Comments:

Now that the authors have discussed in detail the proxies and how they intent to interpret Pb and Nd isotopes in their record, I have a few concerns that I would request them to address. According to what they have written, the following is my understanding

1. Pb isotopes in the water column are primarily controlled by Pb's particle reactive nature. However, according to their own admission (line 75) bottom water control is also present. So, to summarize, Pb isotopes extracted from Fe-Mn oxides coating on bulk sediments are a combination of integrated upper water column and bottom water signature. (In this context upper water column is a loosely used term and has no depth connotation, which might be important especially if one is discussing upper and lower cells of MOC).

Reply #2: Since we cannot clearly draw a line regarding the depth interval covered by the Pb isotope signal of "upper water column" processes we avoid associating it directly with the upper SO circulation cell. This is particularly applicable for deglacial intervals during which the upper SO circulation cell evidently moved south thereby expanding in volume as it moved relative to ODP Site 1094. However, the clear changes seen during the early parts of both Terminations that are independent of the Nd isotopic records suggest that these changes were more or less entirely driven by Pb supply within the upper circulation cell. In the revised version of the manuscript, we either refer to "upper water column processes", or, where the situation was clear, to "upper circulation cell processes".

2. Neodymium isotopes extracted from Fe-Mn oxide coating off bulk sediments, on the other hand is strictly controlled by deep ocean water masses and their mixing proportion, a classical water mass tracer. However, the picture is not that rosy any more as there are reports of benthic flux of Nd perturbing bottom water Nd isotope signatures whereby the bottom signals are altered towards porewater values. This is particularly true when deep ocean circulation is sluggish. Thus, to summarize Nd isotopes (in theory) can be a combination of bottom water mass and porewater compositions.

Reply #3: Agreed. We also already mentioned this in the last version of the manuscript. These properties are introduced in lines 78-84 and further discussed in lines 219-238. We do not regard this porewater control on the early deglacial part of our Nd isotope records as too problematic. Every proxy has its limitations, and given the large amount of research published over the last years using the Nd isotope proxy for paleoceanographic reconstructions by our and other research groups, in the meantime we have a good understanding about key controls over authigenic Nd in deep marine settings¹⁻⁷. The suggestion that authigenic ϵ_{Nd} is seemingly controlled by porewater processes during the early deglacial sections at our site is not per se a negative finding. It rather demonstrates that the deep water advection rates had to be sluggish enough to cause the observed effect. In other words: identifying a porewater signal in our core in the end serves as a paleo-circulation indicator, suggesting a sluggish deep water circulation regime during this time. That said, we still cannot unambiguously exclude the possibility that the very radiogenic early deglacial ϵ_{Nd} may represent a real bottom water ϵ_{Nd} signature (see also reply #12 below).

I agree with the authors that these contrasting characters can work to extract meaningful information especially if the pattern of change of Pb and Nd time series matches, then it would indicate that Pb signals is dominantly bottom water controlled. In the event the pattern of change between Pb and Nd isotope is not coherent, the Pb isotope changes can be interpreted as changes in the upper water column. In my opinion, this argument only works when the influence of porewater on the Nd isotopes can be shown to have minimal to no effect. It is quite clear (radiogenic value ~ -4 that is not common for this part of the ocean) that the downcore Nd record is affected by porewater influence for a prolonged part of their T1 record. The authors in their rebuttal argued that if there has been a prominent oceanic signal, it would have shown in the record and the record would not have been as invariant as they observe. They put forward the stark change in Nd isotopes ~ 13 ka as the evidence that when oceanic processes take over, porewater influence is easily overwhelmed. I understand the line of reasoning that the authors have put forward, however, it is still not as convincing and hard to defend. The authors could acknowledge this point in their manuscript. They should consider incorporating their "reply#8" in the main text so that the readers are aware of the caveat in the interpretation.

Reply #4: The reviewer's concern is reasonable. Given that porewater alteration affected the extracted early deglacial Nd isotope signal, a potential AABW export signal between 18 and

13 ka could in the worst case be missed. However, such a scenario would require that this hypothetical AABW export was very sluggish and not comparable to the modern bottom water situation. Yet the deglacial ϵ_{Nd} signature scatters around of ~ -4 , which is only about 1.5 ϵ_{Nd} units higher than seen at Site MD57-3076 in the South Atlantic during the early deglacial⁸. Even if our record was offset to more radiogenic ϵ_{Nd} by porewater processes, we would expect to see at least some minor changes towards less radiogenic compositions if significant bottom water destratification in the deglacial parts indeed took place. We also would like to reiterate that the onset of AABW export into the northern realm of the Atlantic sector of the SO at ~ 13 ka also coincided with an authigenic Mn concentration spike reported at a nearby site in an earlier study⁹ (see Fig. 2f) and agrees with $\delta^{18}O$ and $\delta^{13}C$ water column reconstructions¹⁰ (see also reply #8 below). Nevertheless, we are happy to follow the reviewer's suggestion to incorporate the content of former "reply#8" in the main text to acknowledge remaining uncertainties contained within our dataset. This information has been added in lines 224-238 of the revised version manuscript. As an additional measure, we slightly amended the title of our manuscript to account for this caveat.

Minor comments:

Line 55: Need a citation.

Reply #5: A reference is added (in line 54)

Line 234: Not really 'several' thousand years.

Reply #6: The "delay" on the order of several thousand years can be more clearly seen in Supplementary Figure 3. Even though there are indeed ϵ_{Nd} variations between 138 and 134ka, the ϵ_{Nd} variations scatter within the error. Thus, we remain convinced that ϵ_{Nd} started to significantly change at 134 ka, while $^{206}Pb/^{204}Pb$ increased as early as 138 ka.

Figure 1: Salinity is unitless.

Reply #7: Figure 1 has been changed to display neutral density as opposed to salinity in order to better define the water masses.

Reviewer #2

Although I no longer have the original, the revised manuscript is easier to read and tells a much more coherent story about changes in the production of Weddell Sea Bottom Water during the last two deglaciations. My only comment this time around would be to encourage the authors to take a look at Sikes et al. (Enhanced $\delta^{13}C$ and $\delta^{18}O$ Differences between the South Atlantic and South Pacific During the Last Glaciation: The Deep Gateway Hypothesis, *Paleoceanography*, 32, <https://doi.org/10.1002/2017PA003118>, 2017). Fig. 3 in the Sikes et al. paper documents changes in $d^{18}O$ and $d^{13}C$ with depth in the South Atlantic and South Pacific during the most recent deglaciation. They find a pattern in the South Atlantic that is much like the one in the present manuscript: an early replacement of water with glacial characteristics above 2 km followed by a later replacement of the deep water below 2 km.

The early replacement in Sikes et al. (2017) takes place during HS1 like the changes in the Pb isotopes in Huang et al., while the late replacement takes place during the YD like Huang et al.'s changes in ϵ_{Nd} . The early upper replacement in Sikes et al. (2017) could easily be described by the southward displacement of major oceanic fronts (lines 120 and 121 in Huang et al.) while the late deep replacement could be explained by a resumption of Weddell Sea Bottom Water formation.

Reply #8: We much appreciate that the reviewer brought the Sikes et al. (2019) study to our attention. Findings presented in this paper indeed support our findings regarding the timing of water column destratification during Termination I. The timing of deep water column changes matches our first report of Weddell Sea derived AABW admixed into LCDW at Site PS1768-8. Given the earlier suggestion of reduced connectivity of the Atlantic and Pacific sector of the SO, the results of Sikes et al. (2017) likely also explain why our ϵ_{Nd} record did not follow the same isotopic evolution than ϵ_{Nd} records in the Pacific sector¹¹. We integrated key findings of Sikes et al. (2017) that apply to our observations in the revised manuscript (line 162-165 and 179-181).

Reviewer #3

In this new version of the manuscript Huang et al. have better detailed the use of Pb isotopes. Since I was not familiar with Pb isotopes, this “new explanation” raises questions with respect to the interpretation of the records.

The Pb record from PS1599, in the Weddell Sea, does not show any time variations, therefore cannot be used to infer changes in Weddell Sea water. The 2 records that are useful are Pb isotopes from PS1768 and Nd isotopes from ODP1094. Both are situated within the CDW. Following the authors' explanations Pb isotopes represent changes occurring over the whole water column, at that location within the Antarctic Circumpolar Current and the CDW, whereas Nd would represent changes in bottom water. I note that both cores are in the deep ocean (3300m and 2800m depth), but not the abyssal ocean. While I agree that Weddell Sea waters could have a significant influence on both records, I doubt that they only record a Weddell Sea signal given the mixing happening within the ACC and CDW.

Reply #9: First, we fear that the reviewer inadvertently mixed up proxies from cores presented in our manuscript. We measured near-Antarctic core PS1599 as a reference for the northern core site aiming to monitor the internal Weddell Sea Nd isotopic evolution. We only presented Nd isotope records from Site PS1599, yet given the comment above we think that the reviewer referred to the invariant Nd isotopic composition of this core (i.e., not a “Pb record”). If our reasoning is correct, then we disagree with the reviewer since this core monitored the southern Weddell Sea Nd isotope composition, which is unsurprisingly invariant since it is located at the Antarctic continental margin. If the Nd isotope composition of water masses in the Weddell Sea had changed significantly over this interval, it should have been recorded at Site PS1599. However, such changes are not seen in this record.

We only measured the coretop Pb isotope composition from Site PS1599 (the data point with highest $^{206}\text{Pb}/^{204}\text{Pb}$ and $^{208}\text{Pb}/^{206}\text{Pb}$ in the green “Western Weddell Sea” area Figure 3a) in the manuscript. The Pb isotope record in PS1599 is not suited to infer general changes in Weddell Sea water because it is dominated by local weathering input due to its ice-proximal position. Ice advance and retreat can incongruently release more radiogenic (high) Pb signal as seen before¹²⁻¹⁴. Presenting a really meaningful integrated Pb isotopic export signal from the Weddell Sea into the SO areas to the north would be a study in its own right. The residence time of dissolved Pb will also be affected by particle density in the water column (biogenic, dust and/or IRD-related), is temporally variable and hard to assess, for which reason we decided to only group coretop Pb isotope compositions as a function of “eastern” or “western” Weddell Sea regions as shown in Figure 3. The ϵ_{Nd} signal is not visibly affected by the local weathering signal, for which reason we present downcore ϵ_{Nd} from Site PS1599 to monitor the Weddell Sea Deep Water signal variability.

Orsi et al.¹⁵ defined the upper water mass boundary of AABW with a neutral density $\gamma^{\text{n}} > 27.27 \text{ kg m}^{-3}$, and Figure 1 has now been revised to display the neutral density allowing the identification of the different water masses in the study area. Hence with regard to the water masses monitored in our two northern sites, today ODP Site 1094 as well as PS1768-8 are both located in the transition zone of AABW sensu stricto and overlying LCDW. However, the reader should keep in mind that prior to its arrival at our core sites, Weddell Sea Deep Water is heavily mixed with LCDW arriving from the Pacific sector of the SO¹⁶, as it is exported from the northern Weddell Sea into the Scotia Sea to the north. We have added this hydrographic information in lines 58-61. Hence while the neutral density close to our sediment sites suggests rather “only” admixtures of AABW to the ambient water mass, the proportion of AABW to ambient water is significant (i.e., Weddell Sea-derived AABW is well mixed into LCDW from the Pacific sector of the SO prior to its arrival at our sites).

Further, it is correct that these two northern cores may not record Weddell Sea-derived AABW if it retreated in the past, but we did not claim that the two cores outside the Weddell Sea only record a Weddell Sea signal. Instead, we wrote in the manuscript: “The two northern sites are ideally located to sensitively record past changes in SO circulation due to their position in the mixing zone between Circumpolar Deep Water (CDW) and Weddell Sea Deep Water”(line 97-100). In order to be more precise we rephrased our text in this context (now lines 95-98) from “Specifically, at ODP Site 1094 and PS1768-8, covariation of Pb and Nd isotopic trends dominantly trace variations in Weddell Sea-derived AABW export” to “Specifically, at ODP Site 1094 and PS1768-8, covariation of Pb and Nd isotopic trends dominantly trace variations in the lower overturning cell”.

As such the Pb record during TI is interpreted as indicating changes in the upper SO cell, which the authors suggest agree with intensified upwelling of CDW. The problem is that the ϵ_{Nd} record does not show any variations until late in the deglaciation (i.e. ~13ka). The authors thus suggest that there was no changes in Weddell-sea derived AABW until that time. This raises several issues:

i) If there was intensified CDW upwelling during the early deglaciation, as also suggested in Figure 4, shouldn't this be visible in the eps(Nd) record?

Reply #10: It depends on the location of the core site. For instance, in the Pacific Sector of Southern Ocean¹¹, the ϵ_{Nd} in the cores (PS75/073-2, PS75/056-1 and PS75/054-1) within the Ross Sea basin started to change at about 18 ka due to the intensified CDW upwelling. However, the ϵ_{Nd} in the PS75/059-2, sitting outside the basin which is similar to our core PS1768-8 in the Atlantic sector, remained stable until about 13 ka. We included a comparison of the ϵ_{Nd} in PS75/059-2 and our core PS1768-8 in Figure 2h and g. See also Reply #8 above in context of comments raised by reviewer #2. To reiterate, Sikes et al (2017) also did not see changes in regional $\delta^{18}O$ and $\delta^{13}C$ in water depths below ~2.5 km water depth until late in the deglaciation.

ii) How can the authors distinguished between Weddell-sea derived AABW and other AABW sources from the eps(Nd) record?

Reply #11: Not every variety of AABW has been probed for its Nd isotopic composition, but we have a generally good understanding of regional differences. Today, Ross Sea AABW within the Pacific sector of the SO has an ϵ_{Nd} of ~-6.6 to -7.2^{17,18}. Weddell Sea Deep Water in the north-western outlet to the Scotia Sea has a composition of ~-9.0¹⁹. This signature is altered to compositions around -8.5 within the Scotia Sea upon mixing with LCDW arriving from the Drake Passage (i.e. the signal advected to Site PS1768-8)¹⁹. Regional AABW formed in the easternmost part of the Weddell Sea (closer to Cape Darnley in the Indian sector of the SO) seems to have an ϵ_{Nd} of around -9.5¹⁹. Finally, Adélie Land Bottom Water ϵ_{Nd} signatures are very close to Weddell Sea Deep Water signatures with a reported value of -8.9²⁰. Orsi et al.¹⁵ argued that Ross Sea AABW is too dense to pass the Drake Passage sill, meaning that Weddell Sea and Ross Sea AABW are not physically connected. Regional AABW varieties arriving from the Indian sector of the SO may have temporally affected relative AABW proportions in the Atlantic sector of the SO close to our core sites, yet these water masses are very similar in their ϵ_{Nd} and if anything should alter ϵ_{Nd} at Site PS1768-8 to less, not more radiogenic compositions in the hypothetical case that these replaced Weddell Sea-derived AABW.

As a bottom line, since we already are at the limit for references in our manuscript we would like to leave a detailed discussion on the Nd isotopic variability of various regional varieties of AABW that do not really affect our interpretation out of the manuscript.

iii) How can this eps(Nd) record, right in the middle of the CDW be so different from other eps(Nd) records from the region: i.e. Piotrowski et al., 2005 and 2012, Skinner et al., 2013 (Geology), and even the CDW in the Pacific side (Basak et al., 2018)? Even though they are a bit northward, both MD07-3076 (Skinner et al., 2013), and TN057-21 (Piotrowski et al., 2012) are quite deep (3800 m and over 4000m depth respectively), and therefore the early eps(Nd) shift is most likely related to AABW changes.

Reply #12: The evolution of the ϵ_{Nd} record of Site PS1768-8 (Figure 2g) is actually very similar to the ϵ_{Nd} data at Site PS75/059-2 (Figure 2h) in the Pacific sector of the SO¹⁹. This finding is striking since the two core sites are located in different sectors of the Southern Ocean and were likely hydrographically not connected¹⁰. As laid out in the manuscript, we interpret this co-variation as the deglacial expansion of Ross Sea- and Weddell Sea-sourced AABW in more northern reaches of the SO.

In the Atlantic sector, the ϵ_{Nd} record at Site PS1768-8 (53.6°S, 4.5°E) is indeed different from the ϵ_{Nd} records at sites MD07-3076 (44. 4°S, 14.1°W) and TN057-21 (41.1°S, 7.9°E) due to their more northern locations. The present-day seawater Nd isotope signatures at 42°S and 53°S are very different¹⁹, and that is why our core PS1768-8 can better trace the AABW export and retreat. At 42°S, a NADW signal can still be observed ($\epsilon_{Nd} = -10.9$)¹⁹, but the seawater ϵ_{Nd} at 53°S is about -8.5 which is very close to the Weddell Sea Deep Water $\epsilon_{Nd} = -9$ ¹⁹. The early deglacial ϵ_{Nd} change at sites MD07-3076 and TN057-21 were interpreted as NADW strengthening but not AABW export in these references. Both sites are located within the Cape Basin, which forms the exit route of NADW into the SO, as already mentioned in the manuscript (lines 187-190). Moreover, the ϵ_{Nd} record in the nearby ODP 1089 at 4620m depth (Figure 2i) also shows no Nd signal variability until 13ka similar to PS1768-8. We cannot unambiguously assess as to why the records from ODP Site 1089²¹ and TNO57-21²² diverge during the deglacial interval. However, both sites acquire modern compositions only during the early Holocene, not before, which agrees with our record. It may indicate that the deglacial interval at Site PS1768-8 is indeed controlled by porewater alteration until the onset of vigorous deep water circulation, yet this is already discussed at length in the manuscript. Nevertheless, we should not dismiss the option that the hydrographic evolution as far south as Site PS1768-8 differed from the more northern sites, and this will need to be tested in future studies.

Despite it being a valuable effort, I thus remain very skeptical about the eps(Nd) record of PS1768, as well as the link between these records and Weddell-sea derived AABW.

Reply #13: We hope our answers above addressed the reviewer's concerns to their satisfaction. If there are further questions, we are also happy to answer.

I also note that from figure 4, it looks like the eps(Nd) record starts to decrease about 1ka, and maybe not "several thousand years" (L.234) after the Pb record, and right during HS11. I am less aware of eps(Nd) record from the region and covering TII, but maybe this makes more sense than the variations recorded during TI.

Reply #14: Please see reply#6 above.

Other points:

- The introduction lacks precision with respect to changes in atmospheric CO₂. For example L. 27: "In either deglacial atm. CO₂ rise scenario" no deglacial atm. CO₂ rise scenario is presented before. To which scenarios are you referring to?

Reply #15: Sentence has been corrected (line 26).

- While I understand that schematics are very valuable, I do not find convincing evidence between what is presented in the manuscript and what is shown in figure 5.

Reply #16: We initially had similar thoughts than reviewer #3 before submission of the manuscript and did not want to include a schematic figure. Before the first submission, we sent our manuscript for “internal review” to several colleagues. They explicitly asked us to add this Figure 4 to help a less involved reader to better follow the discussion. We are open for suggestions and would like to ask the editor and all reviewers for their opinion as to whether we should include this figure or not.

Additional remarks:

Because of the word limit restriction for the length of a *Nature Communications* abstract we removed the last sentence of the previous version of our manuscript (“Overall our data suggest that Weddell Sea AABW export can be reduced or absent during both colder and warmer climates than current.”). Instead we inserted the sentence: “Increasing delivery of Antarctic Pb to regions outside the Weddell Sea traced SO frontal displacements during both glacial terminations.” We have done this change since so far the highly promising properties of using authigenic Pb isotope records in Southern Ocean settings were so far not mentioned at all in the abstract and we feel this is another major outcome of our study.

References cited in replies:

- 1 Abbott, A. N., Haley, B. A. & McManus, J. Bottoms up: Sedimentary control of the deep North Pacific Ocean’s ϵNd signature. *Geology* **43**, 1035-1035 (2015).
- 2 Blaser, P. *et al.* Extracting foraminiferal seawater Nd isotope signatures from bulk deep sea sediment by chemical leaching. *Chemical Geology* **439**, 189-204 (2016).
- 3 Blaser, P. *et al.* The resilience and sensitivity of Northeast Atlantic deep water ϵNd to overprinting by detrital fluxes over the past 30,000 years. *Geochimica et Cosmochimica Acta* **245**, 79-97 (2019).
- 4 Du, J. H., Haley, B. A. & Mix, A. C. Neodymium isotopes in authigenic phases, bottom waters and detrital sediments in the Gulf of Alaska and their implications for paleo-circulation reconstruction. *Geochimica et Cosmochimica Acta* **193**, 14-35 (2016).
- 5 Dausmann, V., Frank, M., Gutjahr, M. & Rickli, J. Glacial reduction of AMOC strength and long-term transition in weathering inputs into the Southern Ocean since the mid-Miocene: Evidence from radiogenic Nd and Hf isotopes. *Paleoceanography* **32**, 265-283 (2017).
- 6 Pöppelmeier, F., Gutjahr, M., Blaser, P., Keigwin, L. D. & Lippold, J. Origin of Abyssal NW Atlantic Water Masses Since the Last Glacial Maximum. *Paleoceanography and Paleoclimatology* **33**, 530-543 (2018).
- 7 Skinner, L. C. *et al.* Rare Earth Elements in early-diagenetic foraminifer 'coatings': Pore-water controls and potential palaeoceanographic applications. *Geochimica Et Cosmochimica Acta* **245**, 118-132 (2019).
- 8 Skinner, L. C. *et al.* North Atlantic versus Southern Ocean contributions to a deglacial surge in deep ocean ventilation. *Geology* **41**, 667-670 (2013).

- 9 Jaccard, S. L., Galbraith, E. D., Martínez-García, A. & Anderson, R. F. Covariation of deep Southern Ocean oxygenation and atmospheric CO₂ through the last ice age. *Nature* **530**, 207 (2016).
- 10 Sikes, E. L., Allen, K. A. & Lund, D. C. Enhanced $\delta^{13}\text{C}$ and $\delta^{18}\text{O}$ Differences Between the South Atlantic and South Pacific During the Last Glaciation: The Deep Gateway Hypothesis. *Paleoceanography* **32**, 1000-1017 (2017).
- 11 Basak, C. *et al.* Breakup of last glacial deep stratification in the South Pacific. *Science* **359**, 900-904, doi:10.1126/science.aao2473 (2018).
- 12 Basak, C. & Martin, E. E. Antarctic weathering and carbonate compensation at the Eocene–Oligocene transition. *Nature Geoscience* **6**, 121 (2013).
- 13 Gutjahr, M., Frank, M., Halliday, A. N. & Keigwin, L. D. Retreat of the Laurentide ice sheet tracked by the isotopic composition of Pb in western North Atlantic seawater during termination 1. *Earth and Planetary Science Letters* **286**, 546-555 (2009).
- 14 Kurzweil, F., Gutjahr, M., Vance, D. & Keigwin, L. Authigenic Pb isotopes from the Laurentian Fan: Changes in chemical weathering and patterns of North American freshwater runoff during the last deglaciation. *Earth and Planetary Science Letters* **299**, 458-465 (2010).
- 15 Orsi, A. H., Johnson, G. C. & Bullister, J. L. Circulation, mixing, and production of Antarctic Bottom Water. *Progress in Oceanography* **43**, 55-109 (1999).
- 16 Naveira Garabato, A. C., McDonagh, E. L., Stevens, D. P., Heywood, K. J. & Sanders, R. J. On the export of Antarctic Bottom Water from the Weddell Sea. *Deep Sea Research Part II: Topical Studies in Oceanography* **49**, 4715-4742 (2002).
- 17 Rickli, J. *et al.* Neodymium and hafnium boundary contributions to seawater along the West Antarctic continental margin. *Earth and Planetary Science Letters* **394**, 99-110 (2014).
- 18 Carter, P., Vance, D., Hillenbrand, C. D., Smith, J. A. & Shoosmith, D. R. The neodymium isotopic composition of waters masses in the eastern Pacific sector of the Southern Ocean. *Geochimica et Cosmochimica Acta* **79**, 41-59 (2012).
- 19 Stichel, T., Frank, M., Rickli, J. & Haley, B. A. The hafnium and neodymium isotope composition of seawater in the Atlantic sector of the Southern Ocean. *Earth and Planetary Science Letters* **317-318**, 282-294 (2012).
- 20 Lambelet, M. *et al.* The Neodymium Isotope Fingerprint of Adelie Coast Bottom Water. *Geophysical Research Letters* **45**, 11247-11256 (2018).
- 21 Lippold, J. *et al.* Deep water provenance and dynamics of the (de)glacial Atlantic meridional overturning circulation. *Earth and Planetary Science Letters* **445**, 68-78 (2016).
- 22 Piotrowski, A. M. *et al.* Reconstructing deglacial North and South Atlantic deep water sourcing using foraminiferal Nd isotopes. *Earth and Planetary Science Letters* **357-358**, 289-297 (2012).

REVIEWERS' COMMENTS:

Reviewer #1 (Remarks to the Author):

I am happy with the response from the authors. I particularly appreciate how the authors have decided to modify the title of the manuscript to accommodate potential caveat. I have no further comments and congratulate the authors on job well done and specially applaud their multiple rounds of careful and detailed revision.

Reviewer #3 (Remarks to the Author):

The authors have adequately answered my concerns. This is a very interesting study that adds important knowledge on deglacial changes in Southern Ocean circulation. This study should motivate further research on changes in the composition of Southern sourced waters.

Replies to the previous version of Nature Communications manuscript (NCOMMS-19-11815C)

No detectable Weddell Sea Antarctic Bottom Water export during the Last and Penultimate Glacial Maximum

by

Huang Huang, Marcus Gutjahr, Anton Eisenhauer and Gerhard Kuhn

Reviewer #1

I am happy with the response from the authors. I particularly appreciate how the authors have decided to modify the title of the manuscript to accommodate potential caveat. I have no further comments and congratulate the authors on job well done and specially applaud their multiple rounds of careful and detailed revision.

Reply #1: We thank reviewer #1 for her/his constructive comments.

Reviewer #3

The authors have adequately answered my concerns. This is a very interesting study that adds important knowledge on deglacial changes in Southern Ocean circulation. This study should motivate further research on changes in the composition of Southern sourced waters.

Reply #2: We thank the reviewer#3 for insightful reviews.